# Predicting the Mechanical Properties of Heat-Treated Woods Using Optimization-Algorithm-Based BPNN

**Runze Zhang and Yujie Zhu ***

College of Engineering and Technology, Northeast Forestry University, Harbin 150040, China; cherish9z@nefu.edu.cn
* Correspondence: zhuyujie004@126.com; Tel.: +86-136-2460-2246

**Abstract:** This paper aims to enhance the accuracy of predicting the mechanical behavior of wood subjected to thermal modification using an improved dung beetle optimization (IDBO) model. The IDBO algorithm improves the original DBO algorithm via three main steps: (1) using piece-wise linear chaotic mapping (PWLCM) to generate the initial dung beetle species and increase its heterogeneity; (2) adopting an adaptive nonlinear decreasing producer ratio model to control the number of producers and boost the algorithm's convergence rate; and (3) applying a dimensional learning-enhanced foraging (DLF) search strategy that optimizes the algorithm's ability to explore and exploit the search space. The IDBO algorithm is evaluated on 14 benchmark functions and outperforms other algorithms. The IDBO algorithm is then applied to optimize a back-propagation (BP) neural network for predicting five mechanical property parameters of heat-treated larch-sawn timber. The results indicate that the IDBO-BP model significantly reduces the error compared with the BP, tent-sparrow search algorithm (TSSA)-BP, grey wolf optimizer (GWO)-BP, nonlinear adaptive grouping grey wolf optimizer (IGWO)-BP and DBO-BP models, demonstrating its superiority in predicting the physical characteristics of lumber after heat treatment.

**Keywords:** dung beetle optimization; BP neural network; wood heat treatment; timber mechanical performance forecast

## 1. Introduction

Timber is a widely utilized material in the construction and furniture industries because it has numerous benefits, such as environmental sustainability, aesthetic appeal and ease of processing. However, its limited stability and durability hinder its application [1,2]. These limitations have prompted the development of various wood modification techniques, such as chemical, physical and biological methods [3]. Heat treatment is a prevalent technique that enhances wood properties by altering its chemical, physical and structural characteristics through exposure to specific temperature and humidity conditions [4]. This treatment increases wood stability, durability and resistance to corrosion and hydrolysis while improving mechanical properties such as strength, stiffness and hardness [5,6]. Common heat treatment methods include vacuum, dry and moist treatments [7].

The improvement of timber properties via heat treatment has been demonstrated by studies. Korkut et al. [8] examined how the thermal process affects red bud maple's surface roughness and mechanical behaviors. The results indicated that increasing temperatures reduce density and moisture content but increase bending strength and surface roughness. Icel et al. [9] demonstrated that heat treatment significantly improves the physical properties, chemical composition and microstructure of spruce and pine, resulting in enhanced stability, durability performance and service life. Xue et al. [10] investigated how high-temperature heat treatment and impregnation modification techniques affect aspen lumber's physical and mechanical characteristics and found significant improvements in mechanical strength and preservation.

Despite its effectiveness in improving wood mechanical properties, heat treatment has certain limitations. Boonstra et al. [11] reported the decomposition of natural wood components during heat treatment, resulting in reduced wood quality. Hill [12] noted that the efficacy of heat treatment is influenced by various factors such as treatment time, temperature, humidity and wood species, making it challenging to control and optimize the process. Goli et al. [1] investigated the impact of heat treatment on the physical and mechanical properties of birch plywood, revealing an increase in density and hardness but a decrease in moisture content and bending strength.

To address these limitations, researchers have explored the use of neural network models to predict wood mechanical properties. Kohonen [13] introduced self-organizing mapping (SOM) as one of the earliest prototypes for applying neural networks to nonlinear prediction problems. C.G.O. [14] highlighted the potential for neural networks to model complex nonlinear relationships for predicting mechanical properties such as strength and stiffness. Adamopoulos et al. [15] investigated the relationship between the fiber properties of recycled pulp and the mechanical properties of corrugated base paper. Multiple linear regression and artificial neural network models were used to predict the tensile strength and compressive strength of corrugated base paper with different fiber sources, and the results showed that the artificial neural network model was more accurate and stable than the multiple linear regression model. You et al. [16] demonstrated that an artificial neural network (ANN) model based on nondestructive vibration testing can successfully predict the MOE of bamboo–wood composites with high accuracy.

Although employing the BP neural network models to forecast the physical characteristics of heat-treated lumber reduces experimental costs, it presents certain challenges, such as susceptibility to local minima during the learning process and a poor generalization ability, resulting in the inaccurate prediction of new data. To address these limitations, some researchers have explored combining BP neural networks with meta-heuristic algorithms to improve prediction accuracy and model robustness. Chen et al. [17] integrated the Aquila Optimization Algorithm (AOA) [18] with BP neural networks to accurately predict the balance water rate and weight ratio of thermal processing timber, and Wang et al. [19] utilized the Carnivorous Plant Algorithm (CPA) [20] to ameliorate BP neural networks for predicting the adhesion intensity and coarseness of the surfaces of heat-treated wood. Their results indicated that both the AOA-BP and CPA-BP models outperform traditional BP neural network models.

Meta-heuristic algorithms can effectively avoid local optima and improve prediction accuracy when combined with BP neural networks. However, local optima may still occur due to inappropriate algorithm parameters or unreasonable algorithm combinations, resulting in poor model performance. To address this issue, some researchers have suggested improving the original meta-heuristic algorithms before applying them to optimize BP neural networks, aiming to increase the model's generalization ability and reliability. For example, Li et al. [21] enhanced the sparrow search algorithm (SSA) [22] with tent chaotic mapping and applied it to optimize BP neural networks for predicting the mechanical characteristics of heat-treated timber. They found that the TSSA-BP model performs well. Ma et al. [23] proposed a nonlinear adaptive grouping strategy for the Gray Wolf Optimization (GWO) [24] algorithm and used it to optimize BP neural networks for timber mechanical performance forecasts. They demonstrated that the proposed IGWO-BP model has much higher prediction accuracy than that of conventional models.

Similarly, the original Dung Beetle Optimization (DBO) [25] algorithm has drawbacks in avoiding local optima and achieving satisfactory algorithmic accuracy for practical engineering applications. To address these flaws, this article proposes an Improved Dung Beetle Optimizer (IDBO) for optimizing BP neural networks. The IDBO algorithm incorporates three main improvements: first, utilizing piece-wise linear chaotic mapping (PWLCM) to initialize the dung beetle population to increase diversity; second, introducing an adaptive parameter adjustment strategy to enhance the early-stage best-finding ability and improve

algorithmic search efficiency; and finally, balancing local and global search capabilities by incorporating a dimensional learning-enhanced foraging strategy (DLF).

The rest of this article is structured as follows: Section 2 introduces the basic theory of BP and DBO; Section 3 presents the IDBO algorithm model; Section 4 verifies the performance of the IDBO algorithm using benchmark functions; Section 5 evaluates the reliability of the suggested IDBO model for wood mechanical property predictions; and Section 6 concludes.

## 2. Theoretical Analysis of the Algorithm

### 2.1. Back-Propagation (BP) Neural Network Models

The BP neural network is a multi-layered feedforward model primarily utilized for supervised learning tasks. It typically includes input, hidden and output layers [26]. Neurons receive inputs from preceding layers and compute a weighted sum that is transformed by an activation function before being output to subsequent layers. The error between the desired and actual outputs is calculated and propagated backward through the network via a back-propagation algorithm. Weights are updated according to each neuron's contribution to the error using the chain rule. Multiple iterations minimize error and enable the network to approximate desired outputs.

This paper used MATLAB's machine learning toolbox (2019a) to create a BP network. Input data included heat treatment temperature, time and relative humidity, and output data comprised Longitudinal Compressive Strength (LCS), Transverse Rupture Strength (TRS), Transverse Modulus of Elasticity (TME), Radial Hardness (RH) and Tangential Hardness (TH). Five separate prediction models were developed using trainlm with a learning rate of 0.01. Figure 1 shows the structure of the BP neural network with a single hidden layer.

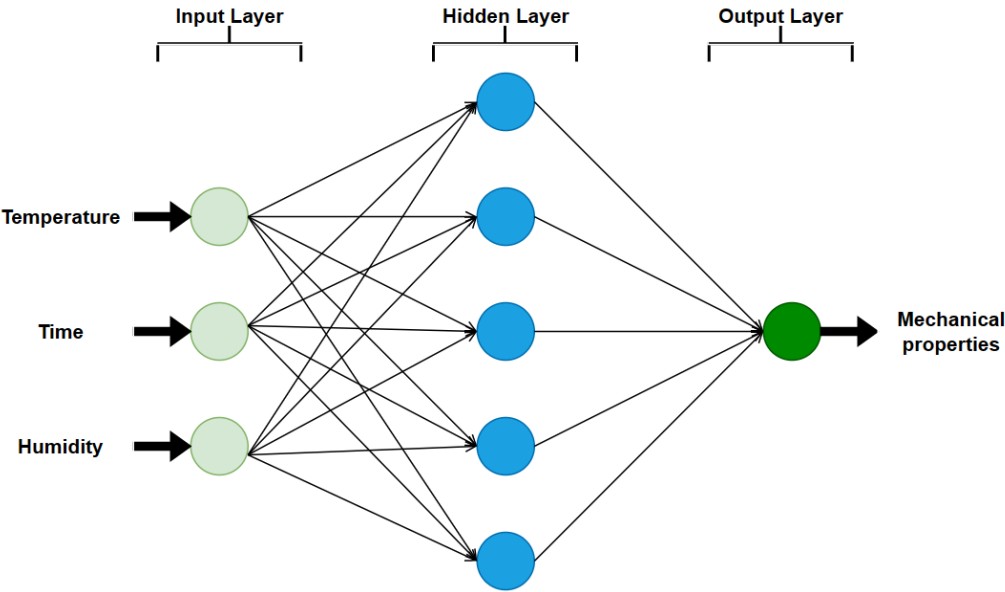

**Figure 1.** BP Network Structure.

### 2.2. The Traditional DBO Algorithm

The DBO is a novel swarm intelligence algorithm that simulates dung beetle habits, such as ball rolling, dancing, foraging, stealing, breeding and other behaviors, and the DBO algorithm comprises four optimization processes: rolling balls, breeding, foraging and stealing [25].

2.2.1. Dung Beetle Ball Rolling

Dung beetle rolling behavior is divided into an obstructed mode and an unobstructed mode.

Obstacle-Free Mode

When the dung beetle is moving forward without obstacles, the dung beetle uses the sun for navigation during dung ball rolling. In this model, the place of the dung beetle alters as the light intensity changes, and the position is renewed, as follows:

$$x_i^{t+1} = x_i^t + a \times k \times x_i^{t-1} + b \times \left| x_i^t - x_{wrost}^t \right| \tag{1}$$

where $t$ indicates the count of the current iterations, and $x_i^t$ is in terms of the position of the $i$th dung beetle in the population at the $t$th permutation. $k \epsilon (0, \ 0.2]$ shows a fixed parameter representing the flexure coefficient, $b$ is an invariant quantity in the range of $(0, 1)$, and $\alpha$ represents a natural coefficient with values of either $-1$ or $1$, with $1$ indicating no deviation and $-1$ indicating deviation from the original direction. $x_{wrost}^t$ means the worst location in the present specie, and the change in light intensity is simulated by $\left| x_i^t - x_{wrost}^t \right|$.

Barrier Mode

The dung beetle, when it encounters an obstacle that prevents it from moving forward, needs to dance to regain a new forward direction. The authors use a tangent function to mimic the dancing behavior as a way to obtain a new rolling direction, which is only considered to be in the range of $[0, \pi]$, and the beetle continues rolling the dung ball once it has determined a new direction. The equation for updating the position at this point is as follows:

$$x_i^{t+1} = x_i^t + tan\theta \left| x_i^t - x_i^{t-1} \right| \tag{2}$$

When $\theta = 0, \frac{\pi}{2}, \pi$, no change occurs in the dung beetle's position.

### 2.2.2. Dung Beetle Breeding

In nature, female dung beetles roll their dung balls to a safe place suitable for egg laying and hide them as a way to provide a suitable habitat for their progeny. Inspired by this, the authors propose a frontier option strategy to model the brood ball location of female dung beetles:

$$\begin{cases} Lf^* = max \left\{ x_{gbest}^t \times (1 - R), Lf \right\} \\ Uf^* = min \left\{ x_{gbest}^t \times (1 + R), Uf \right\} \end{cases} \tag{3}$$

where $R = \frac{1-t}{T_{max}}$, and $T_{max}$ is the upper limit of iterations. The lower and upper limits of the optimization problem are $Lf$ and $Uf$, respectively. The current population attains the global optimum at $x_{gbest}^t$. The authors define the spawning's lower and upper edges region with $Lf$ and $Uf$, which means that the region where the dung beetles spawn is dynamically adjusted with the number of iterations.

When a female dung beetle determines the spawning area, she lays her eggs in that area. Each female dung beetle generates a single brood ball per cycle. The area where oviposition occurs is dynamically adjusted with the count of iterations, so the position of the nestling sphere is also dynamic during the iterations, as defined below:

$$B_i^{t+1} = x_{gbest}^t + b_1 \times \left( B_i^t - Lf^* \right) + b_2 \times \left( B_i^t - Uf^* \right) \tag{4}$$

where $B_i^{t+1}$ is the location of the $i$th brood ball at the $t$th iteration, $b_1$, $b_2$ represent two random and independent vectors that have $D$ components each, and $D$ is the number of parameters in the optimization problem. The position of the nestling ball must be restricted to the spawning area.

### 2.2.3. Dung Beetle Foraging

This behavior is mainly aimed at small dung beetles. Some mature dung beetles emerge from the ground in search of food, and the optimal foraging area for small dung beetles is dynamically updated, as indicated below:

$$\begin{cases} Lf^l = max\{x_{lbest}^t \times (1-R), Lf\} \\ Uf^l = min\{x_{lbest}^t \times (1+R), Uf\} \end{cases} \tag{5}$$

where $R$ is the same as the previous definition, and $x_{lbest}^t$ represents the best local position for the current population. The authors use $Lf^l$ and $Uf^l$ to define the bottom and top boundaries of the foraging region of the small dung beetle, respectively. The equation for updating the position at this point is as follows:

$$x_i^{t+1} = x_i^t + C_1 \times \left(x_i^t - Lf^l\right) + C_2 \times \left(x_i^t - Uf^l\right) \tag{6}$$

where $C_1$ is a number that follows a normal distribution when chosen randomly, namely $C_1 \sim N(0, 1)$, and $C_2$ is a random vector belonging to a range of $(0, 1)$ of $1 \times D$.

### 2.2.4. Dung Beetle Stealing

In the population, there are some dung beetles that steal dung balls from other dung beetles, and the authors update the location of the thieving dung beetles as follows:

$$x_i^{t+1} = x_{lbest}^t + S \times g \times \left(\left|x_i^t - x_{gbest}^t\right| + \left|x_i^t - x_{lbest}^t\right|\right) \tag{7}$$

where $g$ is a vector of dimension $D$ that is randomly chosen, obeying a normal distribution, and $S$ indicates a constant value.

The diagram of the DBO algorithm's process is presented in Figure 2. The algorithm first generates a random initial population of dung beetles in the search space and defines its relevant parameters and then calculates the value of each agent's fitness to adjust their positions based on the objective function, and it finally repeats the above steps until the termination criteria are met, showing the globally optimal solution and its corresponding value of suitability.

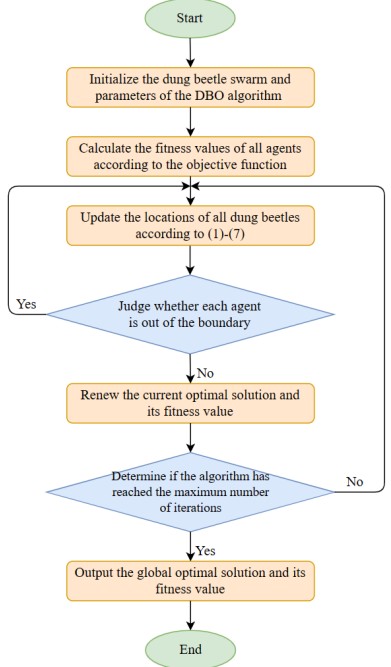

**Figure 2.** Diagram of the DBO algorithm.

## 3. Proposed Method

### 3.1. Improved Dung Beetle Optimizer

Despite its simplicity and successful application to several engineering design problems, the DBO algorithm exhibits limitations, such as poor global searchability and premature convergence to local optima. To address these deficiencies, this paper proposes an improved dung beetle optimizer with specific enhancement strategies.

#### 3.1.1. Piece-Wise Linear Chaotic Mapping

When tackling sophisticated optimization projects, the simple random generation of initial populations by the DBO can result in rapid declines in population diversity and excessive convergence during later iterations. Chaotic sequences have recently been adopted for improving population diversity in meta-heuristic algorithms due to their randomness and ergodicity [27]. The basic approach involves mapping chaotic sequences into individual search spaces using chaos models such as Tent [21], Logistic [28] or Kent [29] chaos mapping.

When selecting a chaotic mapping, two important characteristics—simplicity and ergodicity—must be considered. Segmented linear chaotic mapping satisfies these criteria with its relatively uniform phase distribution and simple equations compared to those of other one-dimensional chaotic systems. This paper uses PWLCM mapping to generate a random sequence with dynamical equations [30] defined as follows:

$$x_{i+1} = F_p(x_i) = \begin{cases} \frac{x_i}{p}, \ 0 \le x_i < p \\ \frac{x_i - p}{0.5 - p}, \ p \le x_i < 0.5 \\ F_p(1 - x_i), \ 0.5 \le x_i < 1 \end{cases} \tag{8}$$

With the control parameter $p\epsilon(0, 0.5)$, the $x_i\epsilon(0, 1)$ system is in a chaotic state. Assigning initial values to the control function $p$ and $x_0$, after circular iterations, a random sequence in the interval $(0, 1)$ can be obtained, which has excellent statistical properties and is commonly applied to generate the initial solution of the algorithm to increase the diversity of the species. When $p = 0.4$, the initial overall (one-dimensional) distribution is as shown in Figure 3.

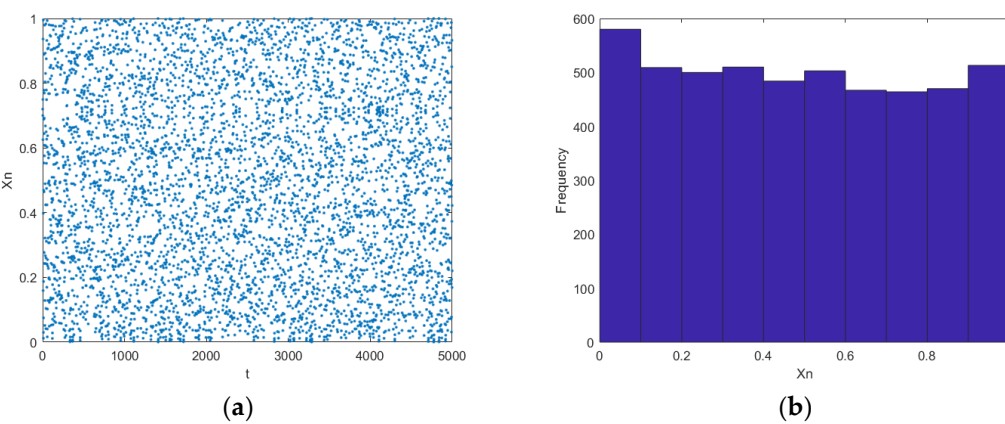

(a)   (b)

**Figure 3.** Population initialization in PWLCM: (**a**) scatter map; (**b**) frequency distribution histogram.

#### 3.1.2. Self-Adaptive Parameter Adjustment Tactics

The DBO algorithm comprises four main components: the global development of producers, egg-laying by female dung beetles, foraging by small dung beetles and stealing behavior by stealing dung beetles. The number of producers determines the explored scope and convergence rate of the algorithm. However, in the original algorithm, the authors do not specify the distribution ratio of these four agents, which may result in incomplete coverage of the search space or slow convergence. To address this issue, this article suggests an adaptively derived non-linear decreasing producer ratio model (Equation (9)) with an

initial producer ratio set to 0.4. With a sufficient number of producers, the algorithm can conduct a more extensive global search during early iterations and fully exploit potential solutions. As iterations progress and the demand for producers decreases, their proportion decreases from 0.4 to 0.2, and exploitation is minimized during the middle and late stages to facilitate rapid convergence.

$$P_{percent} = 0.4 - \frac{t(0.4 - 0.2)}{M} \tag{9}$$

This model enhances algorithm diversity and robustness by dynamically adjusting the number of producers and controlling competition and cooperation among them according to certain strategies. This maintains algorithm versatility while gradually reducing producer numbers during the search process to effectively balance convergence speed and exploration ability. As a result, global exploration capability and convergence speed are improved.

### 3.1.3. Dimension Learning-Enhanced Foraging Search Strategy

During the search process, the DBO algorithm may select a locally optimal solution while ignoring a more optimal global solution due to its random strategy and lack of effective evasion methods. To address this issue, we introduce the Dimension Learning-enhanced Foraging (DLF) search strategy.

In the original DBO algorithm, position updates are obtained according to objective functions corresponding to different agents. This can contribute to slow convergence, trapping in local optima, and the premature loss of population diversity due to random agent selection. In contrast, our proposed DLF search strategy enables agents to update their locations by learning from their neighbors and completing their behaviors accordingly.

In the DLF search strategy, the new location of the dung beetles $X_i(t)$ is obtained from Equation (12), in which the beetle gains information from various neighbors and a randomly chosen agent from the population. Then, in addition to $X_{i-DBO}(t+1)$, the DLF search strategy generates another agent for the new location of beetle $X_i(t)$, named $X_{i-DLF}(t+1)$. To this purpose, first, using Formula (10), the radius $R_i(t)$ is obtained with the magnitude of the displacement vector between the $X_i$ current position $X_i(t)$ and the agent position $X_{i-DBO}(t+1)$.

$$R_i(t) = \|X_i(t) - X_{i-DBO}(t+1)\| \tag{10}$$

Next, the neighborhood of $X_i(t)$ expressed by $N_i(t)$ is derived using Equation (11), which is related to the radius $R_i(t)$, where $D_i$ is the length of the line segment joining $X_i(t)$ and $X_j(t)$.

$$N_i(t) = \{X_j(t) | D_i(X_i(t), X_j(t)) \le R_i(t), X_j(t) \epsilon Pop\} \tag{11}$$

Then, multi-domain learning is performed using Equation (12), where $d$ denotes dimensionality.

$$X_{i-DLF,d}(t+1) = X_{i,d}(t) + rand \times (X_{n,d}(t) - X_{r,d}(t)) \tag{12}$$

Finally, the locations are updated with Formula (13), and the above steps are repeated until a predefined maximum number of iterations is reached and returns the global optimal solution and its corresponding fitness value.

$$X_i(t+1) = \begin{cases} X_{i-DBO}(t+1), & if \ f(X_{i-DBO}) < f(X_{i-DLH}) \\ X_{i-DLF}(t+1) & otherwise \end{cases} \tag{13}$$

### 3.2. The IDBO-BP Algorithm

The BP neural network models are assigned random weights and thresholds with numerous variable parameters that can cause instability in model computation [31]. The

predictive performance of these models can be enhanced by optimizing BP neural networks using DBO. However, the DBO algorithm has issues such as an uneven initial population distribution, susceptibility to local optima and a slow convergence speed. To address these problems, this paper proposes the IDBO algorithm.

First, PWLCM is introduced to initialize the population to produce a more uniform initial solution distribution and high-quality initial solutions while augmenting population richness. Second, using an adaptive parameter adjustment strategy to dynamically tune producer numbers according to the search process accelerates the convergence rate and enhances global exploration capability. Finally, employing the DLF search strategy balances the exploration and exploitation abilities of the algorithm.

The main idea of the IDBO-BP algorithm is to update the weights and thresholds of the BP neural network by continuously updating the positions of the dung beetle swarm until the global best position is found, i.e., the optimal solution.

The diagram of the IDBO-BP algorithm's process is presented in Figure 4. Data are first normalized using Equation (15) before the proportion of dung beetles pushing the ball is dynamically adjusted according to Formula (9). Dung beetle population locations are then initialized using PWLCM mapping, as shown in Equation (8). Fitness values for all dung beetles are calculated, and their locations are updated according to Formulas (1)–(7). The current optimal solution is updated in combination with the DLF strategy before the positions of all dung beetles are updated again in combination with Equation (13). When the iteration limit is reached, the best solution is output along with the optimal parameters of the BP neural network.

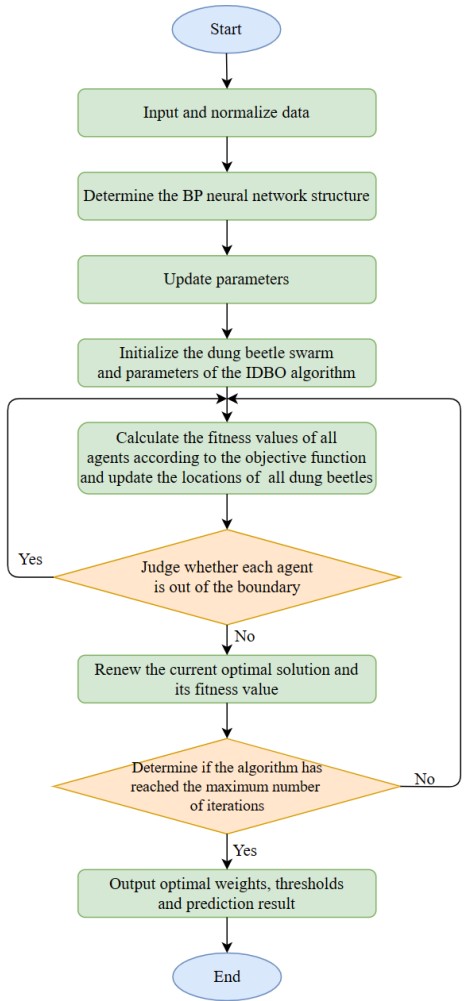

**Figure 4.** Diagram of the IDBO-BP algorithm.

## 4. Evaluate the Effectiveness of the Suggested IDBO Model

The efficacy of the suggested IDBO method is assessed through a series of experiments utilizing various benchmark functions in this section.

### 4.1. Benchmark Functions

To objectively appraise the effectiveness of various meta-heuristic algorithms and to validate the usefulness of the IDBO amelioration strategy, 14 standard test functions were selected from the literature [32], and the CEC2017 test function was utilized to evaluate the capability of the IDBO algorithm. Functions F1–F8 are unimodal with a single global optimal solution and were employed to gauge the velocity and exactness of convergence of the algorithm. Functions F9–F14 are multimodal with a single global optimum and several local optima and were used to estimate the global search and excavation capabilities of the algorithm. The details of these benchmark functions, including their expressions, dimensions, search ranges and theoretical optimal solutions, are given in Tables 1 and 2. To provide a more intuitive understanding of these benchmark functions and their optimal values, Figures 5 and 6 depict 3D views (30 dimensions) of some of these functions.

**Table 1.** Unimodal benchmark functions.

| Function | Dim | Range | $F_{min}$ |
|---|---|---|---|
| $f_1(x) = \sum_{i=1}^{n} x_i^2$ | 30/50/100 | $[-100, 100]$ | 0 |
| $f_2(x) = \sum_{i=1}^{n} \lvert x_i \rvert + \prod_{i=1}^{n} \lvert x_i \rvert$ | 30/50/100 | $[-10, 10]$ | 0 |
| $f_3(x) = \sum_{i=1}^{n} \left( \sum_{j=1}^{i} x_j \right)^2$ | 30/50/100 | $[-100, 100]$ | 0 |
| $f_4(x) = max\{\lvert x_i \rvert, 1 \leq i \leq n\}$ | 30/50/100 | $[-100, 100]$ | 0 |
| $f_5(x) = \sum_{i=1}^{n-1} \left[ 100(x_{i+1} - x_{i^2})^2 + (x_i - 1)^2 \right]$ | 30/50/100 | $[-30, 30]$ | 0 |
| $f_6(x) = \sum_{i=1}^{n} ([x_i + 0.5])^2$ | 30/50/100 | $[-100, 100]$ | 0 |
| $f_7(x) = x_1^2 + 10^6 \sum_{i=2}^{n} x_i^2$ | 30/50/100 | $[-100, 100]$ | 0 |
| $f_8(x) = \sum_{i=1}^{n} x_i^2 + \left( \sum_{i=1}^{n} 0.5 i x_i \right)^2 + \left( \sum_{i=1}^{n} 0.5 i x_i \right)^4$ | 30/50/100 | $[-5, 10]$ | 0 |

**Table 2.** Multimodal benchmark functions.

| Function | Dim | Range | $F_{min}$ |
|---|---|---|---|
| $f_9(x) = \sum_{i=1}^{n} \left[ x_i^2 - 10cos(2\pi x_i) + 10 \right]$ | 30/50/100 | $[-5.12, 5.12]$ | 0 |
| $f_{10}(x) = \sum_{i=1}^{n} \lvert x_i sin(x_i) + 0.1 x_i \rvert$ | 30/50/100 | $[-10, 10]$ | 0 |
| $f_{11}(x) = \frac{\pi}{n} \left\{ 10sin(\pi y_1) + \sum_{i=1}^{n-1} (y_i - 1)^2 \left[ 1 + 10sin^2(\pi y_{i+1}) \right] + (y_n - 1)^2 \right\} + \sum_{i=1}^{n} u(x_i, 10, 100, 4)$, where $y_i = 1 + \frac{x_i+1}{4}$, for all $i = 1, \ldots, n$ $u(x_i, a, k, m) = \begin{cases} k(x_i - a)^m & x_i > a \\ 0 & -a < x_i < a \\ k(-x_i - a)^m & x_i < -a \end{cases}$ | 30/50/100 | $[-50, 50]$ | 0 |
| $f_{12}(x) = 0.1 \{ sin^2(3\pi x_1) + \sum_{i=1}^{n} (x_i - 1)^2 [1 + sin^2(3\pi x_i + 1)] + (x_n - 1)^2 [1 + sin^2(2\pi x_n)] \} + \sum_{i=1}^{n} u(x_i, 5, 100, 4)$ | 30/50/100 | $[-50, 50]$ | 0 |
| $f_{13}(x) = \left[ \frac{1}{n-1} \sum_{i=1}^{n-1} \left( \sqrt{s_i} \times (sin(50 s_i^{0.2}) + 1) \right) \right]^2$ $s_i = \sqrt{x_i^2 + x_{i+1}^2}$ | 30/50/100 | $[-100, 100]$ | 0 |
| $f_{14}(x) = sin^2(\pi y_1) + \sum_{i=1}^{n-1} (y_i - 1)^2 \left[ 1 + 10sin^2(\pi y_i + 1) \right] + (y_n - 1)^2 \left[ 1 + sin^2(2\pi y_n) \right]$, where $y_i = 1 + \frac{x_i-1}{4}$, for all $i = 1, \ldots, n$ | 30/50/100 | $[-10, 10]$ | 0 |

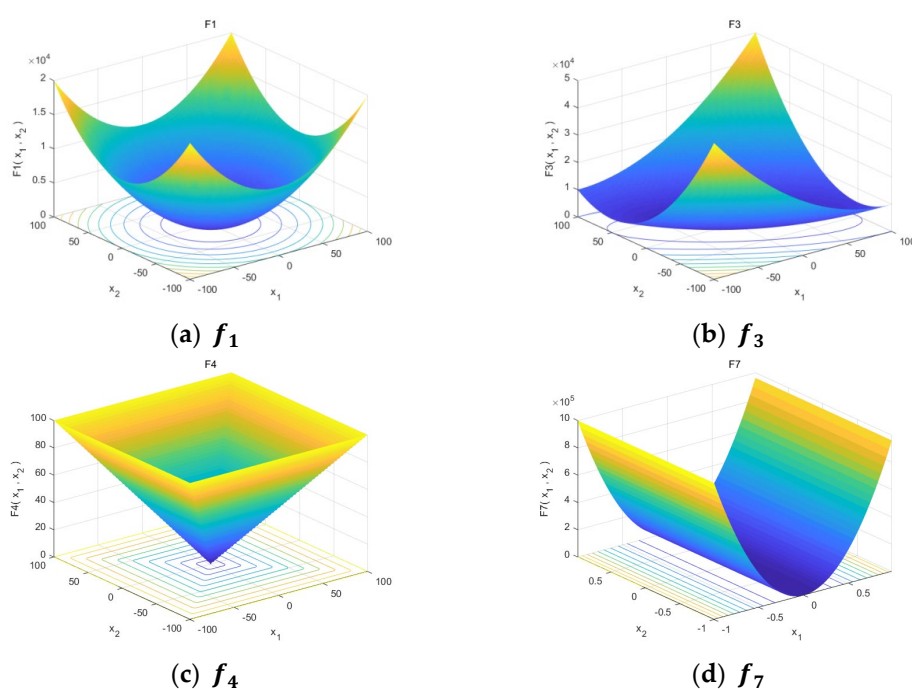

**Figure 5.** Three-dimensional view of partial unimodal test functions.

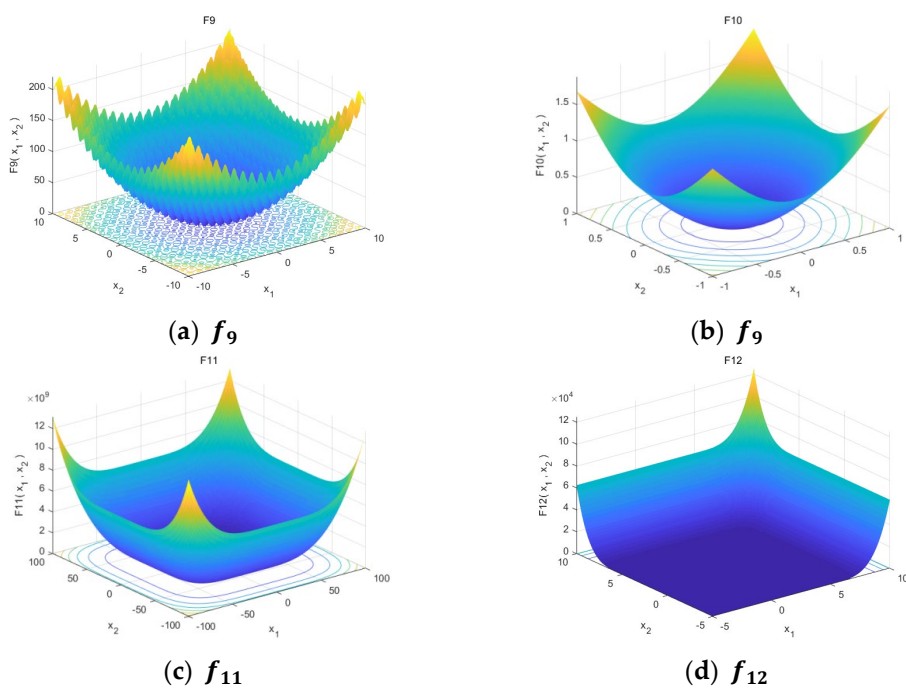

**Figure 6.** Three-dimensional view of partial multimodal test functions.

*4.2. Contrast Algorithm and Experimental Parameter Settings*

To fully validate the reliability of the presented IDBO model, its results were compared with those of four widely used basic metaheuristics: PSO (Eberhart et al., 1995) [33], GWO (Mirjalli et al., 2014), WOA (Mirjalili et al., 2016) [34] and DBO (Xue et al., 2022) [25]. As indicated in Table 3, the parameter settings that were recommended in their respective original works were adopted for the experiments involving these comparison algorithms.

**Table 3.** Algorithms' parameters setting.

| Algorithm | Parameter | Setting |
|---|---|---|
| WOA | $a$ | Gradually reduced from 2 to 0 |
| GWO | $a$ | Uniformly lowered from 2 to 0 |
| PSO | $C_1$ and $C_2$ | 2 |
| | Inertia weight | Linearly decreased from 0.9 to 0.1 |
| DBO | $\alpha$ and $\beta$ | 0.1 |
| | $a$ and $b$ | 0.3 and 0.5 |
| IDBO | $\alpha$ and $\beta$ | 0.1 |
| | $a$ and $b$ | 0.3 and 0.5 |
| | The proportions of the ball-rolling dung beetle, the brood ball, the small dung beetle and the thief were [0.4 0.2], 0.2, 0.2 and [0.2 0.4] | |

To more accurately evaluate the efficacy of the IDBO algorithm and its comparative algorithms, a population size of $N = 30$ was uniformly set, and the upper limit of iterations was fixed at 500. Each model was executed separately 30 times. The dimension $D$ was set to 30, 50 and 100 to examine the effectiveness of the suggested approach in searching for merits across different dimensions. To minimize the influence of randomness in the simulation results, the optimal values, means and standard deviations of the optimization results (fitness) were recorded separately to appraise the exploration performance, accuracy and reliability of the models.

The experiment was implemented on a Windows 11 operating system with an 11th Gen Intel® Core™ i7-11700 processor with 2.5 GHz and 16 GB RAM using MATLAB 2019a for simulation. The optimal fitness, mean fitness and standard error of fitness for the IDBO algorithm and its comparative algorithms are presented in Table 4, where bold values indicate the best consequences. Additionally, the bottom three lines of each table show the 'w/t/l' for the wins (w), ties (t) and losses (l) of each algorithm.

**Table 4.** Unimodal benchmark function optimization results.

| F | D | Index | WOA | GWO | PSO | DBO | IDBO |
|---|---|---|---|---|---|---|---|
| F1 | 30 | Best | $3.94 \times 10^{-83}$ | $7.29 \times 10^{-30}$ | $6.50 \times 10^{2}$ | $1.03 \times 10^{-176}$ | $5.01 \times 10^{-201}$ |
| | | Mean | $7.03 \times 10^{-74}$ | $9.20 \times 10^{-28}$ | $1.77 \times 10^{3}$ | $1.77 \times 10^{-89}$ | $8.31 \times 10^{-151}$ |
| | | STD | $3.41 \times 10^{-73}$ | $9.69 \times 10^{-28}$ | $6.99 \times 10^{2}$ | $9.68 \times 10^{-89}$ | $4.46 \times 10^{-150}$ |
| | 50 | Best | $4.13 \times 10^{-89}$ | $1.29 \times 10^{-29}$ | $6.20 \times 10^{2}$ | $1.71 \times 10^{-163}$ | $1.64 \times 10^{-199}$ |
| | | Mean | $6.15 \times 10^{-72}$ | $9.52 \times 10^{-28}$ | $1.93 \times 10^{3}$ | $1.71 \times 10^{-110}$ | $8.00 \times 10^{-151}$ |
| | | STD | $4.33 \times 10^{-71}$ | $1.13 \times 10^{-27}$ | $8.76 \times 10^{2}$ | $1.21 \times 10^{-109}$ | $5.66 \times 10^{-150}$ |
| | 100 | Best | $3.31 \times 10^{-88}$ | $2.43 \times 10^{-29}$ | $7.52 \times 10^{2}$ | $2.11 \times 10^{-182}$ | $1.89 \times 10^{-208}$ |
| | | Mean | $2.61 \times 10^{-72}$ | $1.20 \times 10^{-27}$ | $2.09 \times 10^{3}$ | $4.94 \times 10^{-105}$ | $6.48 \times 10^{-152}$ |
| | | STD | $2.36 \times 10^{-71}$ | $2.64 \times 10^{-27}$ | $7.95 \times 10^{2}$ | $4.90 \times 10^{-104}$ | $6.48 \times 10^{-151}$ |
| F2 | 30 | Best | $1.71 \times 10^{-57}$ | $1.67 \times 10^{-17}$ | $1.41 \times 10^{1}$ | $7.24 \times 10^{-81}$ | $3.30 \times 10^{-111}$ |
| | | Mean | $8.35 \times 10^{-50}$ | $9.15 \times 10^{-17}$ | $2.03 \times 10^{1}$ | $6.05 \times 10^{-55}$ | $1.98 \times 10^{-83}$ |
| | | STD | $4.56 \times 10^{-49}$ | $5.34 \times 10^{-17}$ | $3.98 \times 10^{0}$ | $3.31 \times 10^{-54}$ | $9.03 \times 10^{-83}$ |
| | 50 | Best | $2.53 \times 10^{-58}$ | $2.48 \times 10^{-17}$ | $9.95 \times 10^{0}$ | $1.67 \times 10^{-83}$ | $1.64 \times 10^{-103}$ |
| | | Mean | $9.04 \times 10^{-51}$ | $9.55 \times 10^{-17}$ | $1.95 \times 10^{1}$ | $2.02 \times 10^{-48}$ | $1.46 \times 10^{-80}$ |
| | | STD | $4.88 \times 10^{-50}$ | $6.54 \times 10^{-17}$ | $4.77 \times 10^{0}$ | $1.43 \times 10^{-47}$ | $1.04 \times 10^{-79}$ |
| | 100 | Best | $5.90 \times 10^{-59}$ | $1.06 \times 10^{-17}$ | $1.05 \times 10^{1}$ | $5.00 \times 10^{-84}$ | $8.80 \times 10^{-107}$ |
| | | Mean | $3.65 \times 10^{-50}$ | $9.57 \times 10^{-17}$ | $1.94 \times 10^{1}$ | $2.17 \times 10^{-56}$ | $1.80 \times 10^{-80}$ |
| | | STD | $3.22 \times 10^{-49}$ | $7.80 \times 10^{-17}$ | $4.20 \times 10^{0}$ | $1.71 \times 10^{-55}$ | $1.45 \times 10^{-79}$ |

**Table 4.** *Cont.*

| F | D | Index | WOA | GWO | PSO | DBO | IDBO |
|---|---|---|---|---|---|---|---|
| F3 | 30 | Best | $8.29 \times 10^3$ | $5.82 \times 10^{-9}$ | $2.23 \times 10^3$ | $1.61 \times 10^{-148}$ | $2.25 \times 10^{-187}$ |
| | | Mean | $4.90 \times 10^4$ | $1.64 \times 10^{-5}$ | $4.87 \times 10^3$ | $5.61 \times 10^{-79}$ | $3.79 \times 10^{-95}$ |
| | | STD | $1.40 \times 10^4$ | $3.50 \times 10^{-5}$ | $1.71 \times 10^3$ | $3.07 \times 10^{-78}$ | $2.08 \times 10^{-94}$ |
| | 50 | Best | $1.48 \times 10^4$ | $2.04 \times 10^{-9}$ | $1.34 \times 10^3$ | $9.33 \times 10^{-144}$ | $6.40 \times 10^{-178}$ |
| | | Mean | $4.36 \times 10^4$ | $3.33 \times 10^{-5}$ | $5.71 \times 10^3$ | $5.63 \times 10^{-81}$ | $4.74 \times 10^{-99}$ |
| | | STD | $1.31 \times 10^4$ | $1.23 \times 10^{-4}$ | $2.02 \times 10^3$ | $2.82 \times 10^{-80}$ | $3.35 \times 10^{-98}$ |
| | 100 | Best | $1.47 \times 10^4$ | $3.82 \times 10^{-9}$ | $2.16 \times 10^3$ | $2.31 \times 10^{-157}$ | $9.16 \times 10^{-183}$ |
| | | Mean | $4.27 \times 10^4$ | $3.43 \times 10^{-5}$ | $5.39 \times 10^3$ | $9.29 \times 10^{-56}$ | $1.63 \times 10^{-85}$ |
| | | STD | $1.26 \times 10^4$ | $1.57 \times 10^{-4}$ | $1.60 \times 10^3$ | $9.29 \times 10^{-55}$ | $1.63 \times 10^{-84}$ |
| F4 | 30 | Best | $2.60 \times 10^{-1}$ | $4.49 \times 10^{-8}$ | $1.79 \times 10^1$ | $2.07 \times 10^{-77}$ | $1.13 \times 10^{-93}$ |
| | | Mean | $5.48 \times 10^1$ | $6.54 \times 10^{-7}$ | $2.85 \times 10^1$ | $3.47 \times 10^{-50}$ | $3.07 \times 10^{-63}$ |
| | | STD | $2.85 \times 10^1$ | $5.12 \times 10^{-7}$ | $6.42 \times 10^0$ | $1.90 \times 10^{-49}$ | $1.68 \times 10^{-62}$ |
| | 50 | Best | $8.90 \times 10^{-1}$ | $2.11 \times 10^{-8}$ | $1.60 \times 10^1$ | $2.22 \times 10^{-85}$ | $5.72 \times 10^{-100}$ |
| | | Mean | $4.97 \times 10^1$ | $7.76 \times 10^{-7}$ | $2.74 \times 10^1$ | $1.01 \times 10^{-51}$ | $1.35 \times 10^{-68}$ |
| | | STD | $2.53 \times 10^1$ | $1.22 \times 10^{-6}$ | $5.21 \times 10^0$ | $7.18 \times 10^{-51}$ | $9.52 \times 10^{-68}$ |
| | 100 | Best | $2.73 \times 10^{-1}$ | $5.70 \times 10^{-8}$ | $1.57 \times 10^1$ | $3.24 \times 10^{-81}$ | $1.78 \times 10^{-100}$ |
| | | Mean | $5.02 \times 10^1$ | $6.89 \times 10^{-7}$ | $2.84 \times 10^1$ | $4.82 \times 10^{-48}$ | $4.69 \times 10^{-66}$ |
| | | STD | $2.65 \times 10^1$ | $8.72 \times 10^{-7}$ | $4.94 \times 10^0$ | $4.38 \times 10^{-47}$ | $4.42 \times 10^{-65}$ |
| F5 | 30 | Best | $2.71 \times 10^1$ | $2.59 \times 10^1$ | $3.58 \times 10^4$ | $2.54 \times 10^1$ | $2.48 \times 10^1$ |
| | | Mean | $2.79 \times 10^1$ | $2.69 \times 10^1$ | $4.52 \times 10^5$ | $2.58 \times 10^1$ | $2.52 \times 10^1$ |
| | | STD | $4.45 \times 10^{-1}$ | $7.15 \times 10^{-1}$ | $4.03 \times 10^5$ | $1.83 \times 10^{-1}$ | $3.06 \times 10^{-1}$ |
| | 50 | Best | $2.72 \times 10^1$ | $2.59 \times 10^1$ | $4.35 \times 10^4$ | $2.52 \times 10^1$ | $2.47 \times 10^1$ |
| | | Mean | $2.82 \times 10^1$ | $2.72 \times 10^1$ | $4.51 \times 10^5$ | $2.58 \times 10^1$ | $2.52 \times 10^1$ |
| | | STD | $4.52 \times 10^{-1}$ | $7.16 \times 10^{-1}$ | $3.86 \times 10^5$ | $2.68 \times 10^{-1}$ | $3.10 \times 10^{-1}$ |
| | 100 | Best | $2.69 \times 10^1$ | $2.53 \times 10^1$ | $3.65 \times 10^4$ | $2.53 \times 10^1$ | $2.47 \times 10^1$ |
| | | Mean | $2.80 \times 10^1$ | $2.70 \times 10^1$ | $4.21 \times 10^5$ | $2.58 \times 10^1$ | $2.52 \times 10^1$ |
| | | STD | $4.45 \times 10^{-1}$ | $7.56 \times 10^{-1}$ | $3.26 \times 10^5$ | $2.17 \times 10^{-1}$ | $2.55 \times 10^{-1}$ |
| | 30 | Best | $0.00 \times 10^0$ | $0.00 \times 10^0$ | $8.05 \times 10^2$ | $0.00 \times 10^0$ | $0.00 \times 10^0$ |
| | | Mean | $0.00 \times 10^0$ | $0.00 \times 10^0$ | $2.56 \times 10^3$ | $0.00 \times 10^0$ | $0.00 \times 10^0$ |
| | | STD | $0.00 \times 10^0$ | $0.00 \times 10^0$ | $8.87 \times 10^2$ | $0.00 \times 10^0$ | $0.00 \times 10^0$ |
| | 50 | Best | $0.00 \times 10^0$ | $0.00 \times 10^0$ | $9.63 \times 10^2$ | $0.00 \times 10^0$ | $0.00 \times 10^0$ |
| | | Mean | $0.00 \times 10^0$ | $0.00 \times 10^0$ | $2.60 \times 10^3$ | $0.00 \times 10^0$ | $0.00 \times 10^0$ |
| | | STD | $0.00 \times 10^0$ | $0.00 \times 10^0$ | $8.53 \times 10^2$ | $0.00 \times 10^0$ | $0.00 \times 10^0$ |
| | 100 | Best | $0.00 \times 10^0$ | $0.00 \times 10^0$ | $8.35 \times 10^2$ | $0.00 \times 10^0$ | $0.00 \times 10^0$ |
| | | Mean | $0.00 \times 10^0$ | $0.00 \times 10^0$ | $2.37 \times 10^3$ | $0.00 \times 10^0$ | $0.00 \times 10^0$ |
| | | STD | $0.00 \times 10^0$ | $0.00 \times 10^0$ | $1.09 \times 10^3$ | $0.00 \times 10^0$ | $0.00 \times 10^0$ |
| F7 | 30 | Best | $3.68 \times 10^{-77}$ | $4.83 \times 10^{-23}$ | $8.52 \times 10^8$ | $1.93 \times 10^{-159}$ | $9.32 \times 10^{-196}$ |
| | | Mean | $1.85 \times 10^{-66}$ | $1.04 \times 10^{-21}$ | $1.75 \times 10^9$ | $6.52 \times 10^{-108}$ | $9.14 \times 10^{-140}$ |
| | | STD | $9.98 \times 10^{-66}$ | $1.42 \times 10^{-21}$ | $5.93 \times 10^8$ | $2.48 \times 10^{-107}$ | $5.01 \times 10^{-139}$ |
| | 50 | Best | $1.87 \times 10^{-81}$ | $8.86 \times 10^{-24}$ | $6.67 \times 10^8$ | $1.04 \times 10^{-158}$ | $8.72 \times 10^{-212}$ |
| | | Mean | $3.32 \times 10^{-66}$ | $7.31 \times 10^{-22}$ | $1.78 \times 10^9$ | $1.48 \times 10^{-105}$ | $3.17 \times 10^{-148}$ |
| | | STD | $2.35 \times 10^{-65}$ | $1.05 \times 10^{-21}$ | $8.39 \times 10^8$ | $9.64 \times 10^{-105}$ | $2.23 \times 10^{-147}$ |
| | 100 | Best | $1.50 \times 10^{-82}$ | $2.06 \times 10^{-23}$ | $3.42 \times 10^8$ | $2.07 \times 10^{-173}$ | $1.84 \times 10^{-200}$ |
| | | Mean | $4.67 \times 10^{-68}$ | $9.61 \times 10^{-22}$ | $1.83 \times 10^9$ | $4.86 \times 10^{-93}$ | $2.34 \times 10^{-141}$ |
| | | STD | $3.28 \times 10^{-67}$ | $1.86 \times 10^{-21}$ | $8.66 \times 10^8$ | $4.86 \times 10^{-92}$ | $1.68 \times 10^{-140}$ |

**Table 4.** *Cont.*

| F | D | Index | WOA | GWO | PSO | DBO | IDBO |
|---|---|---|---|---|---|---|---|
| F8 | 30 | Best | $7.79 \times 10^{-86}$ | $3.58 \times 10^{-31}$ | $6.83 \times 10^{0}$ | $2.68 \times 10^{-173}$ | $2.53 \times 10^{-198}$ |
| | | Mean | $8.40 \times 10^{-77}$ | $2.89 \times 10^{-29}$ | $3.41 \times 10^{1}$ | $3.19 \times 10^{-109}$ | $6.83 \times 10^{-161}$ |
| | | STD | $4.17 \times 10^{-76}$ | $5.64 \times 10^{-29}$ | $1.84 \times 10^{1}$ | $1.68 \times 10^{-108}$ | $2.95 \times 10^{-160}$ |
| | 50 | Best | $1.90 \times 10^{-95}$ | $3.62 \times 10^{-31}$ | $7.56 \times 10^{0}$ | $1.65 \times 10^{-180}$ | $1.59 \times 10^{-215}$ |
| | | Mean | $1.34 \times 10^{-75}$ | $1.93 \times 10^{-29}$ | $2.92 \times 10^{1}$ | $1.24 \times 10^{-116}$ | $3.43 \times 10^{-153}$ |
| | | STD | $5.27 \times 10^{-75}$ | $2.53 \times 10^{-29}$ | $2.23 \times 10^{1}$ | $6.24 \times 10^{-116}$ | $2.42 \times 10^{-152}$ |
| | 100 | Best | $7.99 \times 10^{-93}$ | $2.83 \times 10^{-31}$ | $8.12 \times 10^{0}$ | $2.98 \times 10^{-177}$ | $3.62 \times 10^{-212}$ |
| | | Mean | $1.79 \times 10^{-74}$ | $2.53 \times 10^{-29}$ | $2.90 \times 10^{1}$ | $1.13 \times 10^{-103}$ | $8.64 \times 10^{-153}$ |
| | | STD | $1.29 \times 10^{-73}$ | $4.61 \times 10^{-29}$ | $1.60 \times 10^{1}$ | $1.06 \times 10^{-102}$ | $8.64 \times 10^{-152}$ |
| Rank | 30 | w/t/l | 0/1/7 | 0/1/7 | 0/0/8 | 0/1/7 | **7/1/0** |
| | 50 | w/t/l | 0/1/7 | 0/1/7 | 0/0/8 | 0/1/7 | **7/1/0** |
| | 100 | w/t/l | 0/1/7 | 0/1/7 | 0/0/8 | 0/1/7 | **7/1/0** |

### 4.3. Evaluation of Exploration and Exploitation

The single-peak functions are well-suited to verify the development capability of algorithms in finding optimal solutions. Multimodal functions with numerous locally optimal solutions can assess the ability of IDBO to evade local optima during exploration.

As indicated in Table 4, the IDBO algorithm demonstrates significant improvement for all seven test functions except F6 across all dimensions. Table 5 reveals that the IDBO algorithm outperforms other algorithms in three different dimensions for all five test functions except F13 and that its optimal value, average and standard error are optimal. Thus, it can be inferred that the IDBO algorithm is more effective than DBO in evaluating optimal solutions, which proves that the modification tactic presented in this article can feasibly enhance the original algorithm's ability to explore.

**Table 5.** Multimodal benchmark function optimization results.

| F | D | Index | WOA | GWO | PSO | DBO | IDBO |
|---|---|---|---|---|---|---|---|
| F9 | 30 | Best | $0.00 \times 10^{0}$ | $0.00 \times 10^{0}$ | $7.52 \times 10^{1}$ | $0.00 \times 10^{0}$ | $0.00 \times 10^{0}$ |
| | | Mean | $0.00 \times 10^{0}$ | $2.79 \times 10^{0}$ | $1.09 \times 10^{2}$ | $9.62 \times 10^{-1}$ | $0.00 \times 10^{0}$ |
| | | STD | $0.00 \times 10^{0}$ | $3.25 \times 10^{0}$ | $1.72 \times 10^{1}$ | $3.66 \times 10^{0}$ | $0.00 \times 10^{0}$ |
| | 50 | Best | $0.00 \times 10^{0}$ | $0.00 \times 10^{0}$ | $7.23 \times 10^{1}$ | $0.00 \times 10^{0}$ | $0.00 \times 10^{0}$ |
| | | Mean | $3.41 \times 10^{-15}$ | $7.50 \times 10^{0}$ | $1.13 \times 10^{2}$ | $2.79 \times 10^{-1}$ | $0.00 \times 10^{0}$ |
| | | STD | $1.78 \times 10^{-14}$ | $2.92 \times 10^{1}$ | $1.82 \times 10^{1}$ | $1.27 \times 10^{0}$ | $0.00 \times 10^{0}$ |
| | 100 | Best | $0.00 \times 10^{0}$ | $0.00 \times 10^{0}$ | $6.65 \times 10^{1}$ | $0.00 \times 10^{0}$ | $0.00 \times 10^{0}$ |
| | | Mean | $1.14 \times 10^{-15}$ | $2.47 \times 10^{0}$ | $1.07 \times 10^{2}$ | $3.38 \times 10^{0}$ | $0.00 \times 10^{0}$ |
| | | STD | $8.00 \times 10^{-15}$ | $4.00 \times 10^{0}$ | $1.83 \times 10^{1}$ | $1.68 \times 10^{1}$ | $0.00 \times 10^{0}$ |
| F10 | 30 | Best | $4.32 \times 10^{-58}$ | $1.97 \times 10^{-17}$ | $6.43 \times 10^{0}$ | $4.37 \times 10^{-89}$ | $9.47 \times 10^{-108}$ |
| | | Mean | $3.25 \times 10^{-32}$ | $5.71 \times 10^{-4}$ | $1.08 \times 10^{1}$ | $1.26 \times 10^{-4}$ | $2.44 \times 10^{-81}$ |
| | | STD | $1.78 \times 10^{-31}$ | $7.99 \times 10^{-4}$ | $2.60 \times 10^{0}$ | $3.44 \times 10^{-4}$ | $1.34 \times 10^{-80}$ |
| | 50 | Best | $4.00 \times 10^{-58}$ | $1.75 \times 10^{-16}$ | $3.44 \times 10^{0}$ | $1.99 \times 10^{-88}$ | $2.32 \times 10^{-101}$ |
| | | Mean | $3.83 \times 10^{-1}$ | $4.95 \times 10^{-4}$ | $1.01 \times 10^{1}$ | $1.32 \times 10^{-1}$ | $8.49 \times 10^{-81}$ |
| | | STD | $2.71 \times 10^{0}$ | $5.42 \times 10^{-4}$ | $2.85 \times 10^{0}$ | $9.17 \times 10^{-1}$ | $6.01 \times 10^{-80}$ |
| | 100 | Best | $2.83 \times 10^{-60}$ | $3.32 \times 10^{-17}$ | $4.88 \times 10^{0}$ | $3.72 \times 10^{-87}$ | $1.69 \times 10^{-109}$ |
| | | Mean | $2.28 \times 10^{-1}$ | $4.47 \times 10^{-4}$ | $1.08 \times 10^{1}$ | $1.99 \times 10^{-3}$ | $3.31 \times 10^{-80}$ |
| | | STD | $2.28 \times 10^{0}$ | $5.21 \times 10^{-4}$ | $2.61 \times 10^{0}$ | $1.57 \times 10^{-2}$ | $2.91 \times 10^{-79}$ |

**Table 5.** *Cont.*

| F | D | Index | WOA | GWO | PSO | DBO | IDBO |
|---|---|---|---|---|---|---|---|
| F11 | 30 | Best | $6.35 \times 10^{-3}$ | $1.31 \times 10^{-2}$ | $1.15 \times 10^{1}$ | $1.11 \times 10^{-7}$ | $4.56 \times 10^{-6}$ |
| | | Mean | $2.76 \times 10^{-2}$ | $5.00 \times 10^{-2}$ | $1.07 \times 10^{3}$ | $3.57 \times 10^{-3}$ | $5.73 \times 10^{-5}$ |
| | | STD | $2.02 \times 10^{-2}$ | $2.95 \times 10^{-2}$ | $4.83 \times 10^{3}$ | $1.89 \times 10^{-2}$ | $1.03 \times 10^{-4}$ |
| | 50 | Best | $2.47 \times 10^{-3}$ | $1.22 \times 10^{-2}$ | $1.26 \times 10^{1}$ | $7.74 \times 10^{-8}$ | $3.29 \times 10^{-6}$ |
| | | Mean | $2.34 \times 10^{-2}$ | $4.54 \times 10^{-2}$ | $1.10 \times 10^{3}$ | $2.26 \times 10^{-3}$ | $3.79 \times 10^{-5}$ |
| | | STD | $1.81 \times 10^{-2}$ | $2.56 \times 10^{-2}$ | $3.36 \times 10^{3}$ | $1.47 \times 10^{-2}$ | $4.79 \times 10^{-5}$ |
| | 100 | Best | $3.42 \times 10^{-3}$ | $1.32 \times 10^{-2}$ | $7.66 \times 10^{0}$ | $5.39 \times 10^{-8}$ | $2.17 \times 10^{-6}$ |
| | | Mean | $2.26 \times 10^{-2}$ | $4.46 \times 10^{-2}$ | $1.61 \times 10^{3}$ | $9.76 \times 10^{-5}$ | $7.55 \times 10^{-5}$ |
| | | STD | $1.92 \times 10^{-2}$ | $2.36 \times 10^{-2}$ | $6.56 \times 10^{3}$ | $7.08 \times 10^{-4}$ | $1.96 \times 10^{-4}$ |
| F12 | 30 | Best | $9.44 \times 10^{-2}$ | $3.15 \times 10^{-1}$ | $7.93 \times 10^{2}$ | $1.79 \times 10^{-4}$ | $1.50 \times 10^{-4}$ |
| | | Mean | $5.49 \times 10^{-1}$ | $6.39 \times 10^{-1}$ | $2.14 \times 10^{5}$ | $5.44 \times 10^{-1}$ | $3.32 \times 10^{-2}$ |
| | | STD | $3.20 \times 10^{-1}$ | $1.90 \times 10^{-1}$ | $2.87 \times 10^{5}$ | $4.09 \times 10^{-1}$ | $4.58 \times 10^{-2}$ |
| | 50 | Best | $1.81 \times 10^{-1}$ | $1.00 \times 10^{-1}$ | $6.48 \times 10^{1}$ | $7.70 \times 10^{-4}$ | $5.05 \times 10^{-5}$ |
| | | Mean | $6.09 \times 10^{-1}$ | $6.13 \times 10^{-1}$ | $1.89 \times 10^{5}$ | $6.14 \times 10^{-1}$ | $2.66 \times 10^{-2}$ |
| | | STD | $2.75 \times 10^{-1}$ | $2.44 \times 10^{-1}$ | $3.94 \times 10^{5}$ | $4.19 \times 10^{-1}$ | $4.10 \times 10^{-2}$ |
| | 100 | Best | $1.17 \times 10^{-1}$ | $1.02 \times 10^{-1}$ | $7.83 \times 10^{1}$ | $1.35 \times 10^{-3}$ | $6.25 \times 10^{-5}$ |
| | | Mean | $4.87 \times 10^{-1}$ | $6.46 \times 10^{-1}$ | $3.87 \times 10^{5}$ | $7.15 \times 10^{-1}$ | $3.72 \times 10^{-2}$ |
| | | STD | $2.78 \times 10^{-1}$ | $2.30 \times 10^{-1}$ | $4.63 \times 10^{5}$ | $4.89 \times 10^{-1}$ | $6.20 \times 10^{-2}$ |
| F13 | 30 | Best | $0.00 \times 10^{0}$ | $0.00 \times 10^{0}$ | $0.00 \times 10^{0}$ | $0.00 \times 10^{0}$ | $0.00 \times 10^{0}$ |
| | | Mean | $8.23 \times 10^{-5}$ | $0.00 \times 10^{0}$ | $0.00 \times 10^{0}$ | $0.00 \times 10^{0}$ | $0.00 \times 10^{0}$ |
| | | STD | $3.25 \times 10^{-4}$ | $0.00 \times 10^{0}$ | $0.00 \times 10^{0}$ | $0.00 \times 10^{0}$ | $0.00 \times 10^{0}$ |
| | 50 | Best | $0.00 \times 10^{0}$ | $0.00 \times 10^{0}$ | $0.00 \times 10^{0}$ | $0.00 \times 10^{0}$ | $0.00 \times 10^{0}$ |
| | | Mean | $5.65 \times 10^{-5}$ | $0.00 \times 10^{0}$ | $0.00 \times 10^{0}$ | $0.00 \times 10^{0}$ | $0.00 \times 10^{0}$ |
| | | STD | $3.11 \times 10^{-4}$ | $0.00 \times 10^{0}$ | $0.00 \times 10^{0}$ | $0.00 \times 10^{0}$ | $0.00 \times 10^{0}$ |
| | 100 | Best | $0.00 \times 10^{0}$ | $0.00 \times 10^{0}$ | $0.00 \times 10^{0}$ | $0.00 \times 10^{0}$ | $0.00 \times 10^{0}$ |
| | | Mean | $7.81 \times 10^{-5}$ | $0.00 \times 10^{0}$ | $0.00 \times 10^{0}$ | $0.00 \times 10^{0}$ | $0.00 \times 10^{0}$ |
| | | STD | $3.89 \times 10^{-4}$ | $0.00 \times 10^{0}$ | $0.00 \times 10^{0}$ | $0.00 \times 10^{0}$ | $0.00 \times 10^{0}$ |
| F14 | 30 | Best | $3.69 \times 10^{-1}$ | $8.22 \times 10^{-1}$ | $4.86 \times 10^{2}$ | $8.97 \times 10^{-2}$ | $2.88 \times 10^{-4}$ |
| | | Mean | $9.42 \times 10^{-1}$ | $1.33 \times 10^{0}$ | $1.03 \times 10^{3}$ | $5.50 \times 10^{-1}$ | $8.37 \times 10^{-2}$ |
| | | STD | $4.24 \times 10^{-1}$ | $2.92 \times 10^{-1}$ | $3.21 \times 10^{2}$ | $4.22 \times 10^{-1}$ | $9.28 \times 10^{-2}$ |
| | 50 | Best | $2.55 \times 10^{-1}$ | $6.37 \times 10^{-1}$ | $3.73 \times 10^{2}$ | $2.69 \times 10^{-4}$ | $4.52 \times 10^{-4}$ |
| | | Mean | $9.42 \times 10^{-1}$ | $1.23 \times 10^{0}$ | $9.25 \times 10^{2}$ | $4.83 \times 10^{-1}$ | $1.05 \times 10^{-1}$ |
| | | STD | $3.68 \times 10^{-1}$ | $2.13 \times 10^{-1}$ | $3.46 \times 10^{2}$ | $2.10 \times 10^{-1}$ | $1.11 \times 10^{-1}$ |
| | 100 | Best | $1.93 \times 10^{-1}$ | $8.13 \times 10^{-1}$ | $4.09 \times 10^{2}$ | $1.40 \times 10^{-3}$ | $2.75 \times 10^{-4}$ |
| | | Mean | $8.70 \times 10^{-1}$ | $1.25 \times 10^{0}$ | $9.66 \times 10^{2}$ | $5.16 \times 10^{-1}$ | $7.45 \times 10^{-2}$ |
| | | STD | $3.82 \times 10^{-1}$ | $2.29 \times 10^{-1}$ | $3.04 \times 10^{2}$ | $2.94 \times 10^{-1}$ | $7.85 \times 10^{-2}$ |
| Rank | 30 | w/t/l | 0/1/5 | 0/1/5 | 0/1/5 | 0/1/5 | 4/2/0 |
| | 50 | w/t/l | 0/0/6 | 0/1/5 | 0/1/5 | 0/1/5 | 4/2/0 |
| | 100 | w/t/l | 0/0/6 | 0/1/5 | 0/1/5 | 0/1/5 | 4/2/0 |

*4.4. Evaluation of Convergence Curves*

To more intuitively observe and compare the convergence rate, accuracy and ability of each algorithm to evade local optima, the convergence curves for IDBO and four basic meta-heuristic algorithms $f_1 \sim f_{14}$ (30 dimensions) are presented in Figure 7. The transverse axis represents the number of iterations, whereas the longitudinal axis denotes the order of magnitude of fitness values. Fitness values are expressed as logarithms to base 10 to better illustrate convergence trends.

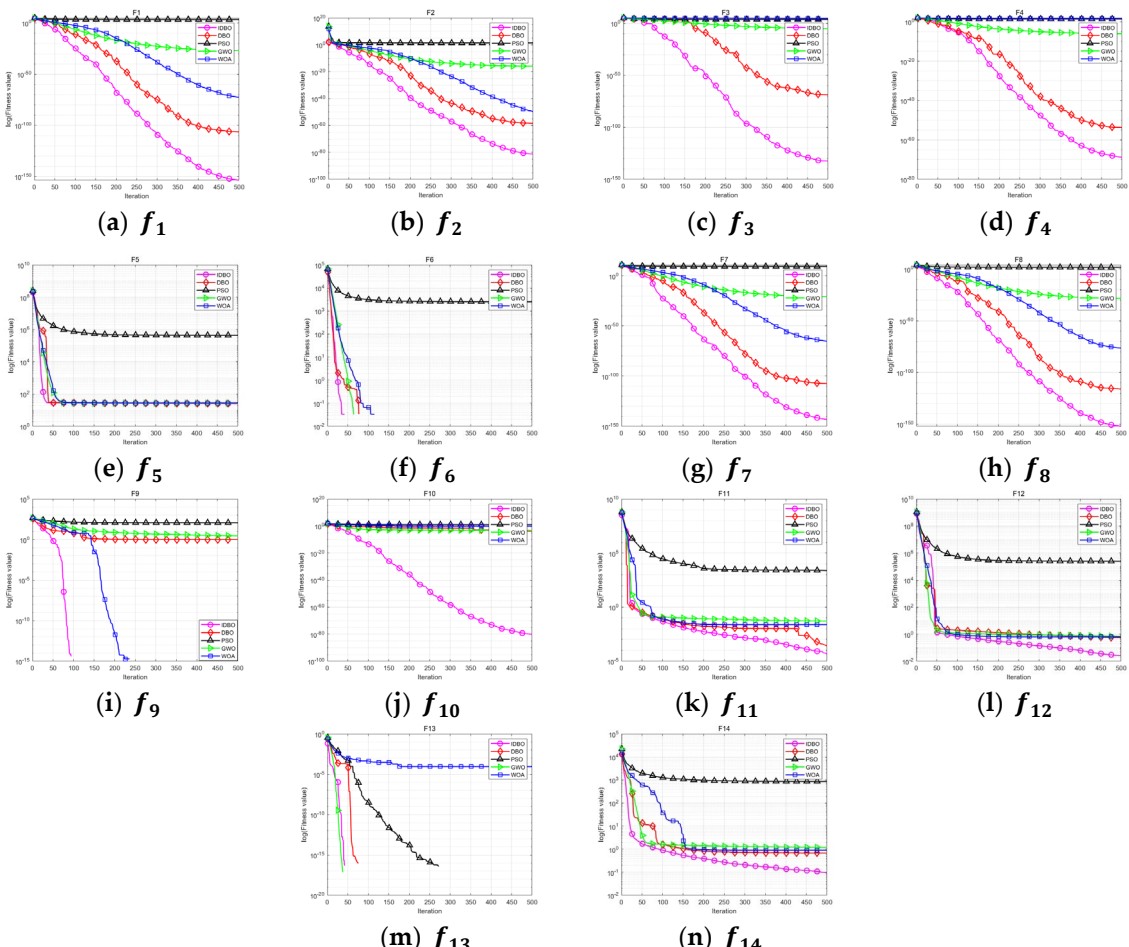

**Figure 7.** Convergence curves of unimodal and multimodal functions.

As shown in Figure 7, IDBO exhibits the fastest convergence and highest accuracy in convergence curves for functions F1–F5, F7–F10 and F14 with a near-linear decrease to theoretical optimal values without stagnation. DBO performs second only to IDBO for these functions and outperforms other algorithms. DBO, GWO and WOA converge to optimal values with minimal stagnation for function F6 but at a slower rate than that of IDBO, and PSO exhibits stagnation at local optima. In the convergence curves for functions F11–F13, IDBO converges rapidly with a quick inflection point to achieve optimal accuracy. This demonstrates that the amelioration method recommended in this paper effectively enhances the original algorithm in terms of convergence rate and accuracy.

### 4.5. Local Optimal Circumvention Evaluation

As previously mentioned, multimodal functions can be used to examine the search behavior of algorithms. As indicated in Table 5, IDBO achieves the best (fitness) optimal values across three different dimensions of 30, 50 and 100 and outperforms other algorithms. This demonstrates that IDBO effectively equilibrates local and global searches to evade local optima. The improvement approach suggested in this article dramatically augments the exploratory potential of the original model.

### 4.6. High-Dimensional Robustness Evaluation

General algorithms may not exhibit robustness and stability when solving complex problem functions in high dimensions, and their ability to find optimal solutions may decrease abruptly. To assess the performance of IDBO in high dimensions, results for IDBO and other algorithms were compared in 50 and 100 dimensions. As presented in Table 4,

for unimodal functions other than F3 and F6, PSO, WOA and GWO all exhibit decreased convergence accuracy in higher dimensions, and DBO and IDBO show decreased accuracy in 50 dimensions but little change in 100 dimensions, indicating stability for both DBO and IDBO at higher dimensions. For function F3, the convergence accuracy of IDBO is 18 orders of scale above that of DBO in 50 dimensions but increases to 29 orders of magnitude higher than that of DBO in 100 dimensions, indicating slightly inferior performance for DBO at higher dimensions.

According to Table 5, for high-dimensional multimodal function F9, the convergence accuracy for WOA and DBO decreases from theoretical optimal values as dimensionality increases, and only IDBO consistently converges to theoretical optimal values with a mean and standard deviation of zero, indicating stable performance for IDBO when seeking high-dimensional multimodal functions. For four test functions, excluding F9 and F13, IDBO's performance at high dimensions is comparable to that at 30 dimensions, achieving optimal mean and standard deviation values. Overall, IDBO exhibits a strong performance when finding optimal solutions for high-dimensional optimization problems, demonstrating its stability and robustness at high dimensions.

Table 6 presents a summary of the performance outcomes for IDBO and other algorithms, as shown in Tables 4 and 5. The total performance metric is employed to calculate TP for each algorithm using Equation (14), in which each algorithm has $Q$ trials and $M$ failed tests.

$$\text{TP} = \left( \frac{Q - M}{Q} \right) \times 100\% \tag{14}$$

**Table 6.** Total performance of IDBO and other basic prevalent meta-heuristics algorithms.

|  | WOA | GWO | PSO | DBO | IDBO |
|---|---|---|---|---|---|
|  | w/t/l | w/t/l | w/t/l | w/t/l | w/t/l |
| D = 30 | 0/2/12 | 0/2/12 | 0/1/13 | 0/2/12 | 11/3/0 |
| D = 50 | 0/1/13 | 0/2/12 | 0/1/13 | 0/2/12 | 11/3/0 |
| D = 100 | 0/1/13 | 0/2/12 | 0/1/13 | 0/2/12 | 11/3/0 |
| Total | 0/4/38 | 0/6/36 | 0/3/39 | 0/6/36 | 33/9/0 |
| TP | 9.52% | 14.29% | 7.14% | 14.29% | 100.00% |

### 4.7. Statistical Analysis

To further evaluate the validity of the suggested enhancement tactics, this paper used a Wilcoxon signed-rank test to compare IDBO with four meta-heuristic algorithms and applied the Friedman test (Equation (15)) to calculate each algorithm's ranking. The number of populations $N = 30$ was set, each test function was subjected to 30 independent runs of each algorithm, dimension $D = 30$, and the Wilcoxon signed-rank test with $\alpha = 0.05$ was implemented for IDBO and other algorithms on 14 test functions. The $\alpha$-values are presented in along with statistics for "}}+", "}}−" and "}} =". "}}+" shows that IDBO clearly outperforms other comparison algorithms, "}}−" indicates inferiority, and "}} =" denotes no significant difference. $N/A$ represents not applicable when both searches for superiority result in 0, indicating a comparable performance. Bold text indicates insignificant or comparable differences.

Table 7 shows that WOA, GWO and DBO have comparable search performances with IDBO for F6, and PSO differs significantly from IDBO. WOA and IDBO have a comparable search performances for F9, and DBO differs insignificantly from IDBO. GWO, PSO and DBO have equivalent search behavior with IDBO for F13, and WOA differs insignificantly from IDBO. GWO, PSO and DBO differ significantly from IDBO for all functions except F6, F9 and F13.

**Table 7.** $\alpha$-values of Wilcoxon signed-rank test.

| Function | WOA | GWO | PSO | DBO |
|---|---|---|---|---|
| F1 | $3.02 \times 10^{-11}$ | $3.02 \times 10^{-11}$ | $3.02 \times 10^{-11}$ | $6.07 \times 10^{-11}$ |
| F2 | $3.02 \times 10^{-11}$ | $3.02 \times 10^{-11}$ | $3.02 \times 10^{-11}$ | $4.20 \times 10^{-10}$ |
| F3 | $3.02 \times 10^{-11}$ | $3.02 \times 10^{-11}$ | $3.02 \times 10^{-11}$ | $2.57 \times 10^{-7}$ |
| F4 | $3.02 \times 10^{-11}$ | $3.02 \times 10^{-11}$ | $3.02 \times 10^{-11}$ | $1.61 \times 10^{-10}$ |
| F5 | $3.02 \times 10^{-11}$ | $3.02 \times 10^{-11}$ | $3.02 \times 10^{-11}$ | $4.18 \times 10^{-9}$ |
| F6 | N/A | N/A | $1.21 \times 10^{-12}$ | N/A |
| F7 | $3.02 \times 10^{-11}$ | $3.02 \times 10^{-11}$ | $3.02 \times 10^{-11}$ | $1.61 \times 10^{-10}$ |
| F8 | $3.02 \times 10^{-11}$ | $3.02 \times 10^{-11}$ | $3.02 \times 10^{-11}$ | $2.61 \times 10^{-10}$ |
| F9 | N/A | $1.16 \times 10^{-12}$ | $1.21 \times 10^{-12}$ | $3.34 \times 10^{-1}$ |
| F10 | $3.02 \times 10^{-11}$ | $3.02 \times 10^{-11}$ | $3.02 \times 10^{-11}$ | $3.69 \times 10^{-11}$ |
| F11 | $3.02 \times 10^{-11}$ | $3.02 \times 10^{-11}$ | $3.02 \times 10^{-11}$ | $1.99 \times 10^{-2}$ |
| F12 | $3.69 \times 10^{-11}$ | $3.02 \times 10^{-11}$ | $3.02 \times 10^{-11}$ | $1.33 \times 10^{-10}$ |
| F13 | $8.15 \times 10^{-2}$ | N/A | N/A | N/A |
| F14 | $3.02 \times 10^{-11}$ | $3.02 \times 10^{-11}$ | $3.02 \times 10^{-11}$ | $1.09 \times 10^{-10}$ |
| +/=/− | 11/3/0 | 12/2/0 | 13/1/0 | 12/2/0 |

Table A2 in Appendix A shows the results of Friedman's test. The IDBO algorithm has a lower average ranking value than that of the other algorithms in all three dimensions, indicating its superior performance. Moreover, Table A2 reveals that the IDBO algorithm's mean value decreases relative to DBO as dimensionality increases. This shows that IDBO is more robust in higher dimensions than DBO and further verifies the effectiveness of our optimization strategy.

$$F_f = \frac{12n}{k(k+1)} \left[ \sum_j R_j^2 - \frac{k(k+1)^2}{4} \right] \tag{15}$$

where $n$ is the count of case tests, $k$ is the quantity of algorithms, and $R_j$ is the mean ranking of the $j$th algorithm.

The IDBO algorithm shows significant improvements in both local and global exploration abilities based on a comprehensive analysis of benchmark function test results, convergence curves, Wilcoxon signed-rank test results, and Friedman test results for each algorithm. It exceeds the original DBO and WOA algorithms and other optimization algorithms that we compare it with in terms of convergence velocity, accuracy and stability. This verifies the performance of the optimization scheme this paper recommends.

## 5. Experimental Research

### 5.1. Data Preprocessing

To ensure an accurate comparison of algorithm results, this paper uses the same data as those used in [35]. The authors used larch-sawn timber of 22 mm thickness from Northeast China. Samples were heat treated at atmospheric pressure with temperature, time and relative humidity as the process parameters. The temperature was divided into five levels (120 °C to 210 °C), time was divided into four levels (0.5 to 3 h), and relative humidity was divided into four levels (0 to 100%). After treatment, specimens were placed at an ambient temperature of (20 ± 2) °C and relative humidity of (65 ± 3)% until reaching a balanced moisture level. Mechanical properties were then measured by GB/T1935-2009 to GB/T1941-2009 standards. For each test, the mean of five replicates was computed, yielding 88 sets of data in total.

To guarantee equity in model comparisons, this article uses the same training and testing samples as those used in [23]. The first 58 samples in Table A1 in Appendix A

formed the training set, and the last 30 samples constituted the testing set. Input data were normalized using Equation (16) to avoid effects on training speed and prediction accuracy.

$$Y_{norm} = \frac{(y - y_{min})}{(y_{max} - y_{min})} \tag{16}$$

$Y_{norm}$ denotes the value of $y$ after scaling it to a unit interval, and $y$ is the original value. The range of $y$ is bounded by $y_{min}$ and $y_{max}$ as the lower and upper limits, respectively.

### 5.2. Model Parameter Setting

The IDBO-BP model was used to forecast the mechanical features and compare the results with those of BP, TSSA-BP, GWO-BP, IGWO-BP and DBO-BP neural networks to demonstrate its prediction capability. The upper limit of the iterations of the BP neural network was set to 1000 with a target error of 0.0001 and a population size of 50.

#### 5.2.1. Selection of Activation Functions

The activation function is a vital component of a neural network that determines how the neurons produce the output from the input. The activation function gives neural networks nonlinear modeling capabilities, allowing them to approximate complex data and functions. The performance and convergence of neural networks depend on the choice of activation function, so selecting a suitable activation function is a critical step in neural network design. Four common activation functions for BP neural network models in MATLAB are LOGSIG, TANSIG, POSLIN and PURELIN.

Table A3 in Appendix A shows the activation function combinations that minimize the error of different models. Table A3 indicates that the optimal activation function combination for the IDBO-BP model is LOGSIG-PURELIN for LCS, obtained via the exhaustive method. Similarly, the optimal activation functions for other models can be derived.

#### 5.2.2. Determination of the Topology

The number of neurons in each layer and the connection between two adjacent layers constitute the topology of a BP neural network model. The topology affects the neural network's complexity and expressiveness, which in turn influence the neural network's performance and convergence. Hence, selecting an appropriate topology is a crucial step in neural network design.

#### Determination of the Number of Neurons in the Hidden Layer

The BP neural network model's structure and performance depend on the number of hidden layer neurons, a key parameter that affects the model's fit to the data. The optimal number of hidden layer neurons should avoid both underfitting and overfitting. Underfitting occurs when the network has too few hidden layer neurons to capture the data's complex features; overfitting or gradient vanishing occurs when the network has too many hidden layer neurons that fit the training data too closely. The number of hidden layer neurons is not fixed but varies according to the problem's complexity and the data's size. Selecting the appropriate number of hidden layer neurons is essential to enhance the model's generalization ability and prediction accuracy.

This paper proposes an empirical formula (Formula (17)) for estimating the number of neurons in the hidden layer as a reference. The number of neurons in the hidden layer varies from 2 to 7. Table A3 in Appendix A shows the neuron configurations that minimize the error of different models. For example, Table A3 indicates that the optimal neuron configuration for the IDBO-BP model for LCS is 2 (single hidden layer), obtained by the trial-and-error method. Similarly, the optimal neuron configuration for other models can be derived.

$$N_h = \frac{N_s}{(\alpha \times (N_i + N_o))}, \alpha \in [2, 7] \tag{17}$$

where $N_h$, $N_i$ and $N_o$ are the number of neurons in the hidden, input and output layers, respectively, and $N_s$ is the number of samples in the training set.

Determination of the Number of Hidden Layers

The hidden layer of a neural network enables it to process non-linearly separable data. Without hidden layers, neural networks can only represent linearly separable functions or decisions. The number of hidden layers and the activation function influence the neural network's representational power and fit. Generally, more hidden layers reduce the error but also increase the network's complexity and training difficulty, and they may cause overfitting. This paper uses neural network models with single and double hidden layers and employs different activation functions to determine the optimal network structure.

Table A3 in Appendix A shows the topologies of the different models at the error minimum. For example, Table A3 indicates that the optimal topology for the IDBO-BP model for TRS is 3-4-6-1, obtained via iterative attempts. Figure 8 shows the corresponding topology schematic diagram. Similarly, the optimal topology for other models can be derived.

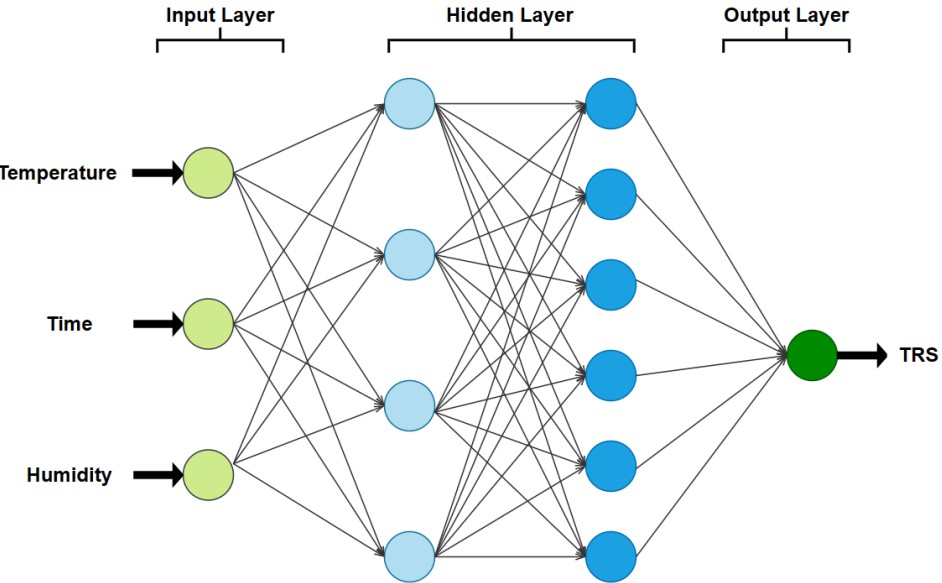

**Figure 8.** Schematic diagram of the IDBO-BP model for the mechanical properties of transverse rupture strength.

*5.3. Model Assessment Standards*

Statistical error is commonly used to evaluate model prediction properties. Common regression evaluation metrics include the mean absolute error (MAE), mean squared error (MSE), mean absolute percentage error (MAPE) and coefficient of determination ($R^2$). MAE reflects the discrepancy between the algorithm's optimal value and the theoretical optimal value. It indicates the algorithm's exploration capability and convergence accuracy. MSE measures the standard error between the predicted and true values, and as the standard error lowers, the model accuracy increases. MAPE measures the relative error between the predicted and true values as a percentage. It is useful for comparing models with different scales of data. R2 measures the model fit to the data, and as it comes closer to 1, the model fit becomes better, and vice versa. As it comes closer to 0, the fit becomes worse. The equations are as follows:

$$\text{MAE} = \frac{1}{N}\sum_{i=1}^{N}|Y_i - Z_i| \qquad (18)$$

$$\text{MSE} = \frac{1}{N}\sum_{i=1}^{N}(Y_i - Z_i)^2 \qquad (19)$$

$$\text{MAPE} = \frac{1}{N}\sum_{i=1}^{N}\left|\frac{Y_i - Z_i}{Y_i}\right| \times 100\% \tag{20}$$

$$R^2 = 1 - \frac{\sum_{i=1}^{n}(Y_i - Z_i)^2}{\sum_{i=1}^{n}\left(Y_i - Y_i\right)^2} \tag{21}$$

where $Y_i$ and $Z_i$ represent the true and predicted values, respectively.

*5.4. Model Performance Comparison Analysis*

To validate the IDBO-BP model, this paper compares it with the BP, TSSA-BP, GWO-BP, IGWO-BP and DBO-BP models. The BP model is the original back-propagation neural network, and TSSA-BP, GWO-BP, IGWO-BP and DBO-BP are optimized versions of BP neural networks that incorporate the TSSA, GWO, IGWO and DBO models, respectively. Table A4 in Appendix A shows the results. For example, using the test set for illustration, IDBO-BP reduces MAE values of LCS, TRS, TME, RH, and TH models by 56%, 31%, 11%, 35% and 31%, respectively, and it reduces the MAPE by 57%, 45%, 38%, 38% and 38%, respectively, compared with the non-optimized BP neural network. For LCS, the significance values of MAE and MAPE of the test set of IDBO-PB corrected with Bonferroni are 0.002 and 0 (less than 0.05), respectively, indicating significant differences between the IDBO-BP and BP models in predicting the mechanical properties of wood. Moreover, compared with the BP, TSSA-PB, GWO-PB, IGWO-PB and DBO-PB models, the IDBO-PB model's predictions are closer to real values, indicating its superior prediction capability. Furthermore, compared with the DBO-PB model, IDBO-PB reduces the MAE values of the testing data of LCS, TRS, TME, RH, and TH models by 43%, 12%, 6%, 7% and 18%, respectively; it reduces the MSE by 78%, 21%, 10%, 8% and 26%, respectively; and it reduces the MAPE by 46%, 10%, 8%, 6% and 21%, respectively. This further verifies the effectiveness of the improved strategy in this paper.

Table A5 in Appendix A shows the rank means and overall rankings of the Friedman tests for the six models based on different evaluation metrics on different parameters. In Friedman's test, the rank mean reflects the solution quality. The rank mean is the average of the solution rank obtained by each algorithm among all the algorithms. As the rank increases, the solution quality increases, and the algorithm performance improves, i.e., it comes closer to the objective function's optimal value. Table A5 shows that IDBO-BP has the highest ranking for all five parameters, outperforming the other models. Figure 9 illustrates the distribution of the six models on their MAE for the LCS test set. The significance value of the null hypothesis for the six models with the same distribution of solutions for the MAE is 0.006, so the null hypothesis is rejected, i.e., there is a significant difference in the solution quality of the six models for MAE, and IDBO-BP has the best performance based on the rank mean.

Figure 10a–e compares the prediction results for five mechanical properties of wood using the IDBO-BP, DBO-BP and BP neural networks with the actual values. The results show that the optimized BP neural networks with the DBO or IDBO models have predictions closer to the true values, indicating that DBO or IDBO improves the BP neural network prediction accuracy. Moreover, the benchmark functions show that the IDBO model performs better than the DBO model in convergence accuracy, stability and exploration capability. This is mainly due to several factors: (1) The IDBO algorithm initializes its dung beetle population using PWLCM chaotic mapping, which enhances population diversity and initial population solution quality. (2) The IDBO algorithm uses an adaptive parameter adjustment strategy with a nonlinear decreasing producer ratio model, which improves searchability in the early and middle stages of the algorithm and increases search range and efficiency. (3) The IDBO algorithm optimizes location updates for small dung beetles by applying a foraging search strategy based on dimensional learning, which balances exploration and exploitation abilities in late iterations.

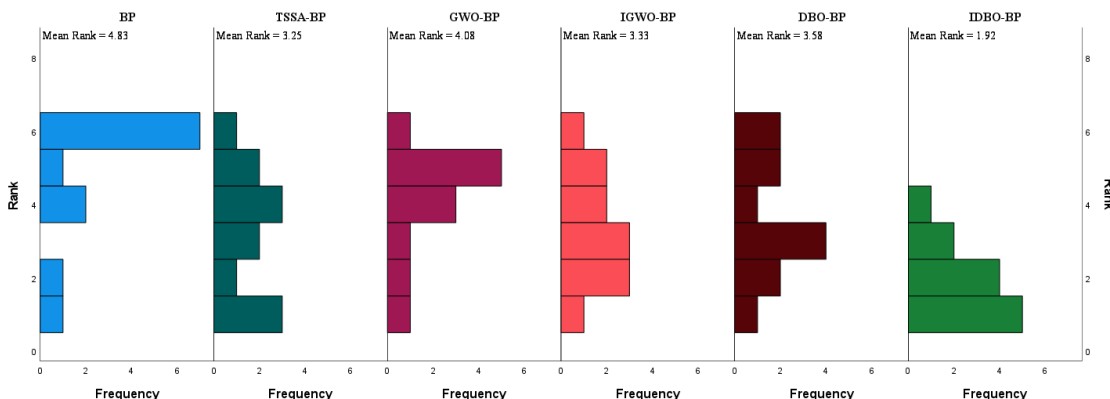

**Figure 9.** Friedman's two-way analysis of variances by ranks for related samples (test sets of LCS).

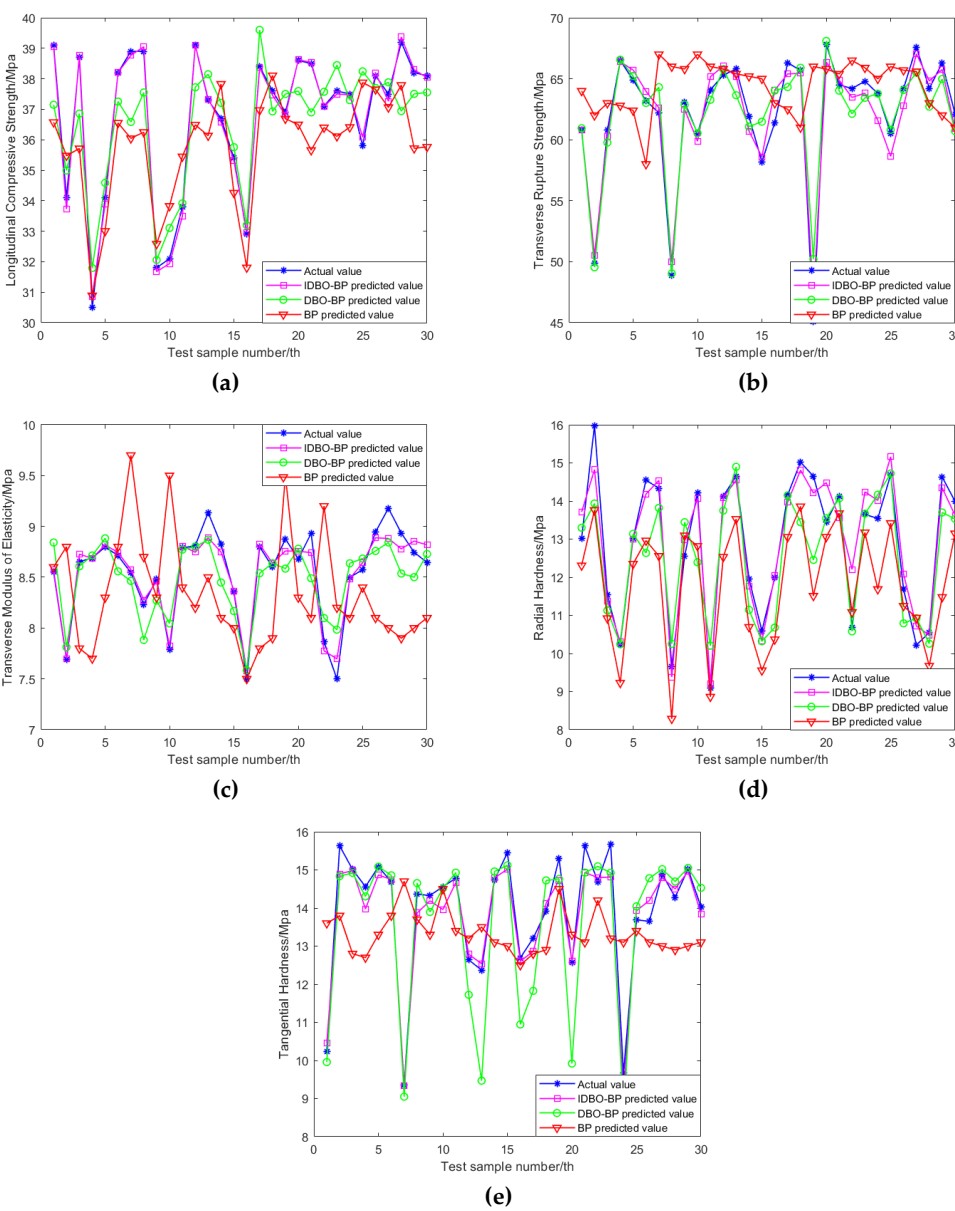

**Figure 10.** Comparison of results for various models based on predictions and actual values: (**a**) longitudinal compressive strength; (**b**) transverse rupture strength; (**c**) transverse modulus of elasticity; (**d**) radial hardness; (**e**) tangential hardness.

In summary, this paper proposes an IDBO algorithm that demonstrates significant improvements in search performance compared to other algorithms, thereby verifying the effectiveness of the enhancement strategy. Additionally, the results demonstrate that the presented IDBO-BP model exhibits outstanding performance in predicting wood mechanical properties.

## 6. Conclusions

- This article proposes the IDBO algorithm to address the limitations of the DBO algorithm. PWLCM mapping is employed to initialize the population and preserve versatility. An adaptive parameter adjustment strategy is introduced to enhance search range and efficiency. Additionally, a DLF strategy is implemented to equilibrium for exploration and exploitation search capabilities, increasing the likelihood of escaping local optima and improving later searchability. The performance of IDBO is evaluated against four basic meta-heuristic algorithms, including DBO, for 14 benchmark functions. The algorithms are ranked by applying the Wilcoxon signed-rank and Friedman tests. The outcomes demonstrate that IDBO outperforms other algorithms in finding solutions for both low- and high-dimensional functions with a single mode or multiple modes, which verifies the effectiveness of the improvement strategy, and it is highly competitive with other meta-heuristics.

- In this paper, five prediction models are separately developed using the IDBO-BP model to predict the LCS, TRS, TME, RH and TH of larch wood after heat treatment with temperature, duration and relative humidity as input variables. The outcomes indicate that the MAE, MSE and MAPE values of the IDBO-BP model are considerably diminished compared with the primitive BP neural network model. The results show that optimizing neural networks model with IDBO significantly improves the prediction accuracy of wood mechanical properties. In addition to comparing the original BP neural network model, this paper also compares it with the TSSA-BP, GWO-BP, IGWO-BP and DBO-BP models. The results denote that the forecast outcomes of the IDBO-BP model are closer to the true values, indicating significant optimization and improved prediction ability.

- This paper compares the optimal prediction models with different parameters and their corresponding topologies and activation functions, and it shows in Table A3 that the same model with different parameters does not necessarily have the same optimal topology. For the LCS, TRS, TME, RH and TH of heat-treated larch wood predicted in this paper, the five most accurate topologies of IDBO-BP that minimize the error are 3-2-1, 3-4-6-1, 3-4-1, 3-5-1 and 3-4-1, respectively.

- The Friedman test can only reflect the quality of the solution, not the diversity of the solution. Therefore, some algorithms may have significant differences in the diversity of solutions, but not in the quality of solutions. The Friedman test is also less robust in some extreme cases; for example, if an algorithm obtains an exceptionally good or bad solution, it may influence the rank and rank mean of other algorithms, thus obscuring the differences between other algorithms. This is illustrated in Table A5. The original BP model ranks first in MSE for both the training and test sets, which may be more susceptible to outlier data because MSE magnifies the prediction error. However, Table A5 also shows that, although the original BP model performs well for MSE, its overall ranking for both the test and training sets is inferior to that of the other five models.

**Author Contributions:** Conceptualization, R.Z.; methodology, R.Z.; software, R.Z.; validation, R.Z.; formal analysis, R.Z.; investigation, R.Z.; resources, R.Z.; data curation, R.Z.; writing—original draft preparation, R.Z.; writing—review and editing, R.Z.; visualization, R.Z.; project administration, R.Z.; funding acquisition, Y.Z. All authors have read and agreed to the published version of the manuscript.

**Funding:** This research was funded by the Fundamental Research Funds for the Central Universities, grant number 2572020BL01, and the Natural Science Foundation of Heilongjiang Province, grant number LH2020C050.

**Data Availability Statement:** From this paper, the data are openly available in a public repository that issues datasets with DOIs. The data that support the findings of this study are openly available in Bioresources at http://doi.org/10.15376/biores.10.3.5758-5776, reference number [35]. The data presented in this study are available in the article.

**Conflicts of Interest:** The authors declare no conflict of interest.

## Appendix A

**Table A1.** Wood treatment conditions and corresponding mechanical properties.

| Test Temperature/°C | Test Time/h | Test Humidity/% | Longitudinal Compressive Strength/MPa | Transverse Rupture Strength/MPa | Transverse Modulus of Elasticity/GPa | Radial Hardness/MPa | Tangential Hardness/MPa |
|---|---|---|---|---|---|---|---|
| 120 | 0.5 | 0 | 41.9 | 67.4 | 9.093 | 14.12 | 15.56 |
| 120 | 0.5 | 40 | 39.8 | 65.3 | 9.038 | 13.02 | 14.69 |
| 120 | 0.5 | 60 | 39.7 | 69.7 | 9.1 | 14.67 | 15.08 |
| 120 | 0.5 | 100 | 39.5 | 67.2 | 8.845 | 14.65 | 15.45 |
| 120 | 1 | 0 | 39.5 | 67.8 | 8.649 | 13.98 | 14.36 |
| 120 | 1 | 40 | 39.5 | 66.4 | 8.752 | 12.98 | 15.59 |
| 120 | 1 | 60 | 39.4 | 67.8 | 9.245 | 13.78 | 15.32 |
| 120 | 1 | 100 | 39.2 | 63.1 | 7.895 | 14.55 | 14.23 |
| 120 | 2 | 0 | 39.2 | 66.9 | 9.074 | 13.33 | 14.23 |
| 120 | 2 | 40 | 39.1 | 68.2 | 8.945 | 12.55 | 14.58 |
| 120 | 2 | 60 | 39.1 | 65.2 | 8.854 | 13.25 | 14.89 |
| 120 | 2 | 100 | 39.1 | 63.2 | 8.933 | 13.36 | 14.56 |
| 120 | 3 | 0 | 39.1 | 66.5 | 8.9 | 13.56 | 14.78 |
| 120 | 3 | 40 | 39.1 | 67.6 | 8.963 | 13.45 | 14.45 |
| 120 | 3 | 60 | 38.9 | 66.6 | 8.745 | 13.01 | 14.69 |
| 120 | 3 | 100 | 38.9 | 64.2 | 8.745 | 12.45 | 14.78 |
| 140 | 0.5 | 0 | 38.9 | 66.7 | 8.978 | 14.69 | 15.56 |
| 140 | 0.5 | 40 | 38.9 | 67.5 | 8.845 | 13.06 | 15.02 |
| 140 | 0.5 | 60 | 38.7 | 66.8 | 9.155 | 14.02 | 14.23 |
| 140 | 0.5 | 100 | 38.7 | 65.3 | 8.877 | 15.02 | 15.01 |
| 140 | 1 | 0 | 38.6 | 66.5 | 9.179 | 14.16 | 15.68 |
| 140 | 1 | 40 | 38.6 | 64.5 | 9.137 | 13.05 | 15.01 |
| 140 | 1 | 60 | 38.5 | 67.2 | 9.024 | 13.49 | 15.17 |
| 140 | 1 | 100 | 38.5 | 63.1 | 8.823 | 13.45 | 15.48 |
| 140 | 2 | 0 | 38.4 | 66.3 | 8.823 | 13.54 | 14.69 |
| 140 | 2 | 40 | 38.4 | 65.7 | 8.852 | 14.69 | 14.58 |
| 140 | 2 | 60 | 38.2 | 67.1 | 8.799 | 13.99 | 14.74 |
| 140 | 2 | 100 | 38.2 | 62.7 | 8.9 | 14.28 | 15.63 |
| 140 | 3 | 0 | 38.2 | 65.4 | 8.811 | 14.39 | 14.23 |
| 140 | 3 | 40 | 38.2 | 64.6 | 8.934 | 13.23 | 14.56 |
| 140 | 3 | 60 | 38.1 | 65.5 | 8.654 | 14.23 | 13.65 |
| 140 | 3 | 100 | 38.1 | 62.1 | 8.798 | 13.56 | 14.02 |
| 160 | 0.5 | 0 | 38.1 | 66.3 | 8.788 | 14.89 | 14.99 |
| 160 | 0.5 | 40 | 38 | 66.9 | 9.011 | 14.87 | 14.36 |
| 160 | 0.5 | 60 | 37.9 | 66.3 | 8.745 | 14.58 | 14.78 |
| 160 | 0.5 | 100 | 37.8 | 65.8 | 8.712 | 14.69 | 15.69 |
| 160 | 1 | 0 | 37.8 | 62.4 | 8.679 | 13.42 | 14.56 |
| 160 | 1 | 40 | 37.6 | 61.4 | 8.645 | 14.09 | 15.3 |
| 160 | 1 | 60 | 37.6 | 62.2 | 8.798 | 14.69 | 15.9 |
| 160 | 1 | 100 | 37.6 | 62.8 | 8.679 | 13.58 | 15.63 |

**Table A1.** *Cont.*

| Test Temperature/°C | Test Time/h | Test Humidity/% | Longitudinal Compressive Strength/MPa | Transverse Rupture Strength/MPa | Transverse Modulus of Elasticity/GPa | Radial Hardness/MPa | Tangential Hardness/MPa |
|---|---|---|---|---|---|---|---|
| 160 | 2 | 0 | 37.6 | 62.2 | 8.727 | 14.63 | 13.92 |
| 160 | 2 | 40 | 37.6 | 62.1 | 8.557 | 14.02 | 14.17 |
| 160 | 2 | 60 | 37.5 | 63.1 | 8.687 | 15.17 | 14.28 |
| 160 | 2 | 100 | 37.5 | 60.9 | 8.611 | 14.65 | 15.09 |
| 160 | 3 | 0 | 37.5 | 61.9 | 8.611 | 13.65 | 14.36 |
| 160 | 3 | 40 | 37.4 | 61.5 | 8.534 | 13.47 | 14.56 |
| 160 | 3 | 60 | 37.3 | 60.8 | 8.601 | 13.58 | 13.89 |
| 160 | 3 | 100 | 37.2 | 60.5 | 8.552 | 13.69 | 14.36 |
| 180 | 0.5 | 0 | 37.2 | 65.9 | 8.601 | 15.21 | 14.03 |
| 180 | 0.5 | 40 | 37.1 | 65.3 | 8.689 | 15.98 | 14.56 |
| 180 | 0.5 | 60 | 37.1 | 66.1 | 8.645 | 16.01 | 13.97 |
| 180 | 0.5 | 100 | 36.9 | 65.7 | 8.599 | 14.32 | 14.33 |
| 180 | 1 | 0 | 36.9 | 65.4 | 8.623 | 15.09 | 13.79 |
| 180 | 1 | 40 | 36.9 | 64.9 | 8.645 | 14.98 | 14.25 |
| 180 | 1 | 60 | 36.7 | 66.3 | 8.579 | 15.45 | 14.08 |
| 180 | 1 | 100 | 36.7 | 64.8 | 8.545 | 14.33 | 13.64 |
| 180 | 2 | 0 | 36.6 | 65.1 | 8.574 | 14.65 | 13.69 |
| 180 | 2 | 40 | 36.5 | 65.8 | 8.6 | 14.13 | 13.59 |
| 180 | 2 | 60 | 36.5 | 64.5 | 8.532 | 13.99 | 14.49 |
| 180 | 2 | 100 | 36.1 | 64.2 | 8.544 | 15.1 | 13.54 |
| 180 | 3 | 0 | 36 | 64.1 | 8.6 | 14.21 | 14.06 |
| 180 | 3 | 40 | 35.9 | 64.2 | 8.541 | 13.99 | 14.21 |
| 180 | 3 | 60 | 35.8 | 64.8 | 8.456 | 14.58 | 13.98 |
| 180 | 3 | 100 | 35.8 | 63.8 | 8.499 | 14.99 | 13.69 |
| 200 | 0.5 | 0 | 35.8 | 62.1 | 8.483 | 12 | 13.6 |
| 200 | 0.5 | 40 | 35.5 | 60.6 | 8.475 | 11.96 | 12.99 |
| 200 | 0.5 | 60 | 35.4 | 59.9 | 8.399 | 11.45 | 13.21 |
| 200 | 1 | 0 | 35.4 | 61.9 | 8.422 | 11.69 | 12.98 |
| 200 | 1 | 40 | 35.1 | 60.8 | 8.489 | 11.46 | 12.64 |
| 200 | 1 | 60 | 34.6 | 61.2 | 8.321 | 11.54 | 12.35 |
| 200 | 2 | 0 | 34.5 | 61.2 | 8.369 | 11.99 | 13.02 |
| 200 | 2 | 40 | 34.5 | 60.8 | 8.354 | 11.15 | 12.69 |
| 200 | 2 | 60 | 34.2 | 60.5 | 8.211 | 10.65 | 12.49 |
| 200 | 3 | 0 | 34.1 | 60.9 | 8.249 | 10.68 | 12.73 |
| 200 | 3 | 40 | 34.1 | 59.8 | 8.231 | 11.05 | 12.57 |
| 200 | 3 | 60 | 34.1 | 58.2 | 8.011 | 10.22 | 12.37 |
| 210 | 0.5 | 0 | 33.9 | 50.1 | 7.856 | 10.23 | 10.98 |
| 210 | 0.5 | 40 | 33.8 | 50.8 | 7.789 | 10.59 | 9.98 |
| 210 | 0.5 | 60 | 33.2 | 49.9 | 7.865 | 10.55 | 10.23 |
| 210 | 1 | 0 | 32.9 | 50.6 | 7.765 | 10.21 | 10.65 |
| 210 | 1 | 40 | 32.9 | 49.8 | 7.712 | 9.98 | 10.21 |
| 210 | 1 | 60 | 32.8 | 48.9 | 7.498 | 10.01 | 10.65 |
| 210 | 2 | 0 | 32.5 | 49.1 | 7.689 | 9.98 | 9.64 |
| 210 | 2 | 40 | 32.1 | 49.5 | 7.712 | 9.65 | 9.35 |
| 210 | 2 | 60 | 31.8 | 49.6 | 7.623 | 10.03 | 9.67 |
| 210 | 3 | 0 | 31.5 | 47.8 | 7.5 | 9.21 | 8.91 |
| 210 | 3 | 40 | 30.8 | 46.5 | 7.412 | 9.1 | 8.21 |
| 210 | 3 | 60 | 30.5 | 45.1 | 7.321 | 9.03 | 8.99 |

**Table A2.** Order by Friedman test in dimensions $D$ = 30, 50 and 100.

| Alg. | D | F1 | F2 | F3 | F4 | F5 | F6 | F7 | F8 | F9 | F10 | F11 | F12 | F13 | F14 | Avg. Rank | Overall Rank |
|---|---|---|---|---|---|---|---|---|---|---|---|---|---|---|---|---|---|
| | 30 | 3 | 3 | 5 | 4.67 | 3.67 | 2.5 | 3 | 3 | 1.83 | 2.33 | 3 | 3 | 4.33 | 3.33 | 3.26 | 3 |
| WOA | 50 | 3 | 2.33 | 5 | 4.67 | 3.67 | 2.5 | 3 | 3 | 2.17 | 3.67 | 3 | 3 | 4.33 | 3.33 | 3.33 | 3 |
| | 100 | 3 | 3 | 5 | 4.67 | 3.67 | 2.5 | 3 | 3 | 2.17 | 3.67 | 3 | 3 | 4.33 | 3.33 | 3.38 | 4 |
| | 30 | 4 | 4 | 3 | 3 | 3.33 | 2.5 | 4 | 4 | 3.17 | 4 | 4 | 3.33 | 2.67 | 3.33 | 3.45 | 4 |
| GWO | 50 | 4 | 4 | 3 | 3 | 3.33 | 2.5 | 4 | 4 | 3.83 | 2.67 | 4 | 2.67 | 2.67 | 3.67 | 3.38 | 4 |
| | 100 | 4 | 4 | 3 | 3 | 3.33 | 2.5 | 4 | 4 | 2.83 | 2.67 | 4 | 2.67 | 2.67 | 3.33 | 3.29 | 3 |
| | 30 | 5 | 5 | 4 | 4.33 | 5 | 5 | 5 | 5 | 5 | 5 | 5 | 5 | 2.67 | 5 | 4.71 | 5 |
| PSO | 50 | 5 | 5 | 4 | 4.33 | 5 | 5 | 5 | 5 | 4.67 | 5 | 5 | 5 | 2.67 | 5 | 4.69 | 5 |
| | 100 | 5 | 5 | 4 | 4.33 | 5 | 5 | 5 | 5 | 5 | 5 | 5 | 5 | 2.67 | 5 | 4.71 | 5 |
| | 30 | 2 | 2 | 2 | 2 | 1.67 | 2.5 | 2 | 2 | 3.17 | 2.67 | 1.67 | 2.67 | 2.67 | 2.33 | 2.24 | 2 |
| DBO | 50 | 2 | 2.67 | 2 | 2 | 1.67 | 2.5 | 2 | 2 | 2.83 | 2.67 | 1.67 | 3.33 | 2.67 | 1.67 | 2.26 | 2 |
| | 100 | 2 | 2 | 2 | 2 | 1.67 | 2.5 | 2 | 2 | 3.5 | 2.67 | 1.67 | 3.33 | 2.67 | 2.33 | 2.31 | 2 |
| | 30 | 1 | 1 | 1 | 1 | 1.33 | 2.5 | 1 | 1 | 1.83 | 1 | 1.33 | 1 | 2.67 | 1 | **1.33** | **1** |
| IDBO | 50 | 1 | 1 | 1 | 1 | 1.33 | 2.5 | 1 | 1 | 1.5 | 1 | 1.33 | 1 | 2.67 | 1.33 | **1.33** | **1** |
| | 100 | 1 | 1 | 1 | 1 | 1.33 | 2.5 | 1 | 1 | 1.5 | 1 | 1.33 | 1 | 2.67 | 1 | **1.31** | **1** |

Table A3. Optimal prediction models with different parameters and the corresponding topologies and activation functions.

| Parms | Model | Neuron Configuration | Topology | Hidden and Output Activations | Train | | | | Test | | | |
|---|---|---|---|---|---|---|---|---|---|---|---|---|
| | | | | | MAE | MSE | MAPE | $R^2$ | MAE | MSE | MAPE | $R^2$ |
| LCS | BP | 2 | 3-2-1 | LOGSIG-PURELIN | 0.2206 | 0.0020 | 0.9335% | 0.9770 | 0.1945 | 0.0011 | 0.5332% | 0.9868 |
| | GWO-BP | 3 | 3-3-1 | LOGSIG-PURELIN | 0.1956 | 0.1054 | 0.5263% | 0.9785 | 0.1372 | 0.0724 | 0.4919% | 0.9940 |
| | TSSA-BP | 4 | 3-4-1 | LOGSIG-TANSIG | 0.1592 | 0.1282 | 0.4342% | 0.9759 | 0.1481 | 0.0642 | 0.4270% | 0.9887 |
| | DBO-BP | (3, 7) | 3-3-7-1 | LOGSIG-PURELIN | 0.1412 | 0.1052 | 0.3786% | 0.9830 | 0.1503 | 0.0566 | 0.4260% | 0.9879 |
| | IGWO-BP | (2, 2) | 3-2-2-1 | PURELIN-LOGSIG | 0.1652 | 0.1200 | 0.4449% | 0.9805 | 0.1291 | 0.0345 | 0.3591% | 0.9907 |
| | IDBO-BP | 2 | 3-2-1 | LOGSIG-PURELIN | 0.1315 | 0.0996 | 0.3513% | 0.9834 | 0.0856 | 0.0122 | 0.2286% | 0.9978 |
| TRS | BP | 2 | 3-2-1 | LOGSIG-PURELIN | 1.2157 | 0.0379 | 2.4030% | 0.9363 | 1.1994 | 0.0753 | 2.4487% | 0.9275 |
| | TSSA-BP | 2 | 3-2-1 | LOGSIG-PURELIN | 1.2191 | 2.2895 | 1.9534% | 0.9236 | 1.2609 | 2.2655 | 2.0291% | 0.9430 |
| | GWO-BP | (3, 7) | 3-3-7-1 | LOGSIG-PURELIN | 1.1097 | 1.8158 | 1.7920% | 0.9433 | 0.9837 | 1.9907 | 1.5331% | 0.9272 |
| | IGWO-BP | 3 | 3-3-1 | LOGSIG-TANSIG | 1.1020 | 2.2158 | 1.8002% | 0.9310 | 0.9519 | 1.5641 | 1.5302% | 0.9557 |
| | DBO-BP | 7 | 3-7-1 | POSLIN-PURELIN | 1.0349 | 1.7647 | 1.6610% | 0.9510 | 0.9345 | 1.4363 | 1.4881% | 0.9484 |
| | IDBO-BP | (4, 6) | 3-4-6-1 | TANSIG-PURELIN | 0.9032 | 1.4304 | 1.4750% | 0.9601 | 0.8218 | 1.1362 | 1.3392% | 0.9683 |
| TME | BP | 3 | 3-3-1 | LOGSIG-PURELIN | 0.1710 | 0.0006 | 2.2422% | 0.8260 | 0.0928 | 0.0005 | 1.4955% | 0.8929 |
| | GWO-BP | (3, 7) | 3-3-7-1 | LOGSIG-PURELIN | 0.0925 | 0.0306 | 1.0968% | 0.8638 | 0.1010 | 0.0164 | 1.1576% | 0.9002 |
| | TSSA-BP | (2, 2) | 3-2-2-1 | LOGSIG-PURELIN | 0.1207 | 0.0500 | 2.0647% | 0.7992 | 0.0977 | 0.0158 | 1.1322% | 0.7225 |
| | IGWO-BP | 3 | 3-3-1 | LOGSIG-PURELIN | 0.1144 | 0.0324 | 1.3611% | 0.8424 | 0.0923 | 0.0155 | 1.0797% | 0.9018 |
| | DBO-BP | (3, 7) | 3-3-7-1 | LOGSIG-PURELIN | 0.1024 | 0.0273 | 1.2126% | 0.8658 | 0.0879 | 0.0127 | 1.0182% | 0.9053 |
| | IDBO-BP | 4 | 3-4-1 | POSLIN-PURELIN | 0.0849 | 0.0266 | 1.0132% | 0.8743 | 0.0824 | 0.0115 | 0.9340% | 0.9156 |
| RH | BP | (5, 6) | 3-5-6-1 | POSLIN-PURELIN | 0.4763 | 0.0049 | 3.5198% | 0.9249 | 0.4963 | 0.0078 | 3.8911% | 0.8986 |
| | TSSA-BP | (3, 7) | 3-3-7-1 | LOGSIG-PURELIN | 0.4205 | 0.3300 | 4.1028% | 0.8901 | 0.3919 | 0.3644 | 3.8402% | 0.9023 |
| | GWO-BP | 5 | 3-5-1 | POSLIN-PURELIN | 0.4009 | 0.2375 | 2.9805% | 0.9233 | 0.5008 | 0.3834 | 3.8098% | 0.8785 |
| | IGWO-BP | (4, 6) | 3-4-6-1 | TANSIG-PURELIN | 0.4256 | 0.2896 | 3.2035% | 0.9107 | 0.4095 | 0.2953 | 3.0822% | 0.8975 |
| | DBO-BP | 2 | 3-2-1 | TANSIG-PURELIN | 0.3928 | 0.2466 | 2.9423% | 0.8985 | 0.3478 | 0.2053 | 2.5616% | 0.9285 |
| | IDBO-BP | 5 | 3-5-1 | POSLIN-PURELIN | 0.3353 | 0.2122 | 2.5198% | 0.9372 | 0.3236 | 0.1889 | 2.4020% | 0.9458 |
| TH | BP | (7, 5) | 3-7-5-1 | LOGSIG-LOGSIG | 0.3850 | 0.0024 | 2.8660% | 0.9412 | 0.4308 | 0.0073 | 3.4057% | 0.8996 |
| | GWO-BP | 2 | 3-2-1 | LOGSIG-PURELIN | 0.2857 | 0.2283 | 2.8154% | 0.9615 | 0.3720 | 0.2421 | 3.0862% | 0.8625 |
| | TSSA-BP | (4, 6) | 3-4-6-1 | TANSIG-PURELIN | 0.3743 | 0.2175 | 2.8032% | 0.9354 | 0.3839 | 0.2201 | 2.9423% | 0.8896 |
| | IGWO-BP | 4 | 3-4-1 | TANSIG-PURELIN | 0.3156 | 0.1671 | 2.3664% | 0.9566 | 0.3824 | 0.2373 | 2.9295% | 0.8899 |
| | DBO-BP | (7, 5) | 3-7-5-1 | LOGSIG-LOGSIG | 0.3335 | 0.1700 | 2.4652% | 0.9532 | 0.3604 | 0.2092 | 2.6622% | 0.8996 |
| | IDBO-BP | 4 | 3-4-1 | TANSIG-PURELIN | 0.2611 | 0.1249 | 1.9515% | 0.9676 | 0.2962 | 0.1544 | 2.1062% | 0.9399 |

Table A4. Pairwise comparisons of the prediction performance of different models with IDBO-BP.

| Parms | Sample1-Sample 2 | Train | | | | | | | | Test | | | | | | | |
|---|---|---|---|---|---|---|---|---|---|---|---|---|---|---|---|---|---|
| | | MAE | Adj. Sig.[a] | MSE | Adj. Sig.[a] | MAPE | Adj. Sig.[a] | $R^2$ | Adj. Sig.[a] | MAE | Adj. Sig.[a] | MSE | Adj. Sig.[a] | MAPE | Adj. Sig.[a] | $R^2$ | Adj. Sig.[a] |
| LCS | IDBO-BP-DBO-BP | 6.85% | 0.743 | 5.32% | 1 | 7.22% | 0.003 | −0.05% | 0.051 | 43.07% | 0.436 | 78.44% | 0.743 | 46.35% | 0.007 | −1.00% | 0.743 |
| | IDBO-BP-IGWO-BP | 20.40% | 1 | 17.05% | 1 | 21.05% | 0.011 | −0.30% | 0.269 | 33.71% | 0.954 | 64.60% | 0.954 | 36.35% | 0.007 | −0.71% | 0.954 |
| | IDBO-BP-GWO-BP | 32.76% | 0.572 | 5.52% | 1 | 33.25% | 0.001 | −0.50% | 1 | 37.65% | 0.068 | 83.14% | 0.132 | 53.53% | 0.001 | −0.38% | 0.011 |
| | IDBO-BP-TSSA-BP | 17.38% | 1 | 22.32% | 1 | 19.10% | 0.096 | −0.77% | 0.945 | 42.21% | 1 | 80.98% | 1 | 46.47% | 0.016 | −0.92% | 0.572 |
| | IDBO-BP-BP | 40.38% | 0.005 | −4858.66% | 0.068 | 62.37% | 0 | −0.66% | 0.103 | 56.00% | 0.002 | −1016.77% | 0.329 | 57.13% | 0 | −1.11% | 0.002 |
| TRS | IDBO-BP-DBO-BP | 12.73% | 0.44 | 18.94% | 1 | 11.20% | 1 | −0.96% | 0.556 | 12.06% | 0.428 | 20.89% | 1 | 10.01% | 1 | −2.10% | 0.185 |
| | IDBO-BP-IGWO-BP | 18.04% | 0.269 | 35.45% | 1 | 18.06% | 1 | −3.13% | 0.269 | 13.67% | 0.945 | 27.36% | 1 | 12.48% | 1 | −1.32% | 0.638 |
| | IDBO-BP-GWO-BP | 18.61% | 0.945 | 21.22% | 1 | 17.69% | 1 | −1.79% | 1 | 16.46% | 0.061 | 42.92% | 1 | 12.65% | 1 | −4.43% | 0.02 |
| | IDBO-BP-TSSA-BP | 25.92% | 0.035 | 37.52% | 1 | 24.49% | 1 | −3.96% | 0.945 | 34.83% | 0.035 | 49.85% | 1 | 34.00% | 0.572 | −2.68% | 1 |
| | IDBO-BP-BP | 25.71% | 0.001 | −3676.83% | 0.001 | 38.62% | 0.096 | −2.55% | 0.103 | 31.48% | 0.006 | −1409.50% | 0 | 45.31% | 0.002 | −4.41% | 0.061 |
| TME | IDBO-BP-DBO-BP | 17.08% | 0.794 | 2.37% | 1 | 16.44% | 0.164 | −0.98% | 0.393 | 6.20% | 0.132 | 9.71% | 0.572 | 8.27% | 0.005 | −1.13% | 1 |
| | IDBO-BP-IGWO-BP | 25.75% | 0.269 | 17.97% | 1 | 25.56% | 0.42 | −3.79% | 0.269 | 10.66% | 0.572 | 25.86% | 1 | 13.49% | 0.954 | −1.53% | 1 |
| | IDBO-BP-GWO-BP | 8.13% | 0.945 | 13.07% | 1 | 7.62% | 0.42 | −1.22% | 1 | 18.36% | 0.002 | 29.85% | 0.034 | 19.31% | 0.181 | −1.71% | 0.007 |
| | IDBO-BP-TSSA-BP | 29.62% | 0.035 | 46.76% | 1 | 50.93% | 0.035 | −9.40% | 0.945 | 15.60% | 0.096 | 27.14% | 0.246 | 17.50% | 0.181 | −26.72% | 0.181 |
| | IDBO-BP-BP | 50.33% | 0.001 | −4103.20% | 0.023 | 54.81% | 0.001 | −5.86% | 0.103 | 11.15% | 0.181 | −2041.22% | 0.436 | 37.55% | 0 | −2.54% | 0.068 |
| RH | IDBO-BP-DBO-BP | 14.63% | 0.361 | 13.98% | 1 | 14.36% | 1 | −4.30% | 0.185 | 6.97% | 1.000 | 8.00% | 1 | 6.23% | 1 | −1.86% | 0.119 |
| | IDBO-BP-IGWO-BP | 21.21% | 0.269 | 26.72% | 1 | 21.34% | 1 | −2.90% | 0.269 | 20.99% | 1 | 36.03% | 1 | 22.07% | 1 | −5.38% | 0.638 |
| | IDBO-BP-GWO-BP | 16.36% | 0.945 | 10.65% | 1 | 15.46% | 1 | −1.50% | 1 | 35.40% | 1 | 50.74% | 1 | 36.95% | 1 | −7.67% | 0.02 |
| | IDBO-BP-TSSA-BP | 20.25% | 0.035 | 35.70% | 0.572 | 38.58% | 1 | −5.29% | 0.945 | 17.44% | 0.096 | 48.18% | 0.954 | 37.45% | 0.572 | −4.82% | 1 |
| | IDBO-BP-BP | 29.60% | 0.001 | −4200.16% | 0.016 | 28.41% | 0.048 | −1.32% | 0.103 | 34.80% | 1.000 | −2308.90% | 0.005 | 38.27% | 0.011 | −5.26% | 0.061 |
| TH | IDBO-BP-DBO-BP | 21.71% | 0.7 | 26.50% | 1 | 20.84% | 0.087 | −1.50% | 0.556 | 17.83% | 0.366 | 26.17% | 1 | 20.88% | 0.151 | −4.48% | 0.113 |
| | IDBO-BP-IGWO-BP | 17.27% | 0.269 | 25.22% | 1 | 17.53% | 0.42 | −1.14% | 0.269 | 22.56% | 0.945 | 34.92% | 1 | 28.10% | 1 | −5.62% | 0.638 |
| | IDBO-BP-GWO-BP | 8.62% | 0.945 | 45.27% | 1 | 30.68% | 0.42 | −0.63% | 1 | 20.39% | 0.061 | 36.20% | 1 | 31.75% | 0.035 | −8.97% | 0.02 |
| | IDBO-BP-TSSA-BP | 30.25% | 0.035 | 42.55% | 1 | 30.38% | 0.035 | −3.44% | 0.945 | 22.87% | 0.035 | 29.84% | 1 | 28.42% | 0.061 | −5.64% | 1 |
| | IDBO-BP-BP | 32.18% | 0.001 | −5117.57% | 1 | 31.91% | 0.001 | −2.80% | 0.103 | 31.25% | 0.006 | −2013.85% | 1 | 38.16% | 0 | −4.48% | 0.061 |

Each row compares sample 1 with sample 2 and calculates the percentage by which sample 1 reduces the error based on sample 2 (a negative sign indicates the percentage by which $R^2$ increases). Asymptotic significances (2-sided tests) are displayed. The significance level is 0.050. [a] Significance values have been adjusted with the Bonferroni correction for multiple tests.

**Table A5.** Model ranking by parameters and evaluation metrics with Friedman test.

| Parms | Model | Train | | | | Test | | | | Avg. Rank | | Overall Rank | |
|---|---|---|---|---|---|---|---|---|---|---|---|---|---|
| | | MAE | MSE | MAPE | $R^2$ | MAE | MSE | MAPE | $R^2$ | Include MSE | Exclude MSE | Include MSE | Exclude MSE |
| LCS | BP | 5 | 1 | 5.42 | 2.17 | 4.83 | 1 | 5.08 | 2.25 | 2.24 | 2.65 | 5 | 6 |
| | TSSA-BP | 2.67 | 4 | 3.08 | 3.92 | 3.25 | 4.08 | 3.5 | 3.58 | 1.64 | 0.83 | 2 | 2 |
| | GWO-BP | 3.83 | 4.5 | 4.08 | 3.33 | 4.08 | 4.75 | 4.08 | 2.58 | 2.43 | 1.69 | 6 | 5 |
| | IGWO-BP | 3.5 | 4.42 | 3.58 | 3.33 | 3.33 | 4.17 | 3.67 | 3.75 | 1.95 | 1.17 | 3 | 3 |
| | DBO-BP | 3.75 | 3.92 | 3.83 | 3.67 | 3.58 | 4.25 | 3.67 | 3.67 | 1.96 | 1.25 | 4 | 4 |
| | IDBO-BP | 2.25 | 3.17 | 1 | 4.58 | 1.92 | 2.75 | 1 | 5.17 | 0.29 | −0.60 | 1 | 1 |
| TRS | BP | 3.25 | 1 | 5.17 | 3.25 | 4 | 1 | 5.08 | 2.92 | 1.67 | 1.89 | 4 | 6 |
| | TSSA-BP | 4.08 | 4.33 | 3.5 | 3 | 3.92 | 4.25 | 3.75 | 3.33 | 2.19 | 1.49 | 6 | 5 |
| | GWO-BP | 4.17 | 4.25 | 3.83 | 3.17 | 3.58 | 4.08 | 3.42 | 3.58 | 2.07 | 1.38 | 5 | 4 |
| | IGWO-BP | 2.83 | 3.75 | 2.67 | 4.33 | 3.5 | 4 | 3.25 | 3 | 1.58 | 0.82 | 2 | 2 |
| | DBO-BP | 3.17 | 3.58 | 2.75 | 3.67 | 3.5 | 3.92 | 3.33 | 3.42 | 1.65 | 0.94 | 3 | 3 |
| | IDBO-BP | 3.5 | 4.08 | 3.08 | 3.58 | 2.5 | 3.75 | 2.17 | 4.75 | 1.34 | 0.49 | 1 | 1 |
| TME | BP | 4.08 | 1 | 4.75 | 3.83 | 4.67 | 1 | 5.08 | 2.83 | 1.74 | 1.99 | 3 | 6 |
| | TSSA-BP | 3.58 | 3.75 | 3.17 | 3.67 | 3.83 | 4.5 | 3.5 | 3.08 | 1.95 | 1.22 | 4 | 3 |
| | GWO-BP | 3.17 | 4.67 | 3.67 | 2.5 | 3.67 | 5 | 4.33 | 2.33 | 2.46 | 1.67 | 6 | 5 |
| | IGWO-BP | 3.25 | 4 | 3 | 3.75 | 3.33 | 3.58 | 3 | 4 | 1.55 | 0.81 | 2 | 2 |
| | DBO-BP | 3.75 | 4.17 | 3.5 | 3.25 | 3.75 | 4.25 | 3.5 | 3.75 | 1.99 | 1.25 | 5 | 4 |
| | IDBO-BP | 3.17 | 3.42 | 2.92 | 4 | 1.75 | 2.67 | 1.58 | 5 | 0.81 | 0.07 | 1 | 1 |
| RH | BP | 4.58 | 1 | 5.08 | 3.42 | 5.17 | 1 | 5.25 | 3.75 | 1.86 | 2.15 | 4 | 6 |
| | TSSA-BP | 3.17 | 5.08 | 3.75 | 2.33 | 3 | 5.17 | 4.25 | 2.08 | 2.50 | 1.63 | 6 | 5 |
| | GWO-BP | 3.5 | 3.83 | 3.08 | 3.42 | 2.92 | 3.5 | 3 | 3.58 | 1.60 | 0.92 | 3 | 3 |
| | IGWO-BP | 3.5 | 4.08 | 3.25 | 3.58 | 3.92 | 4.25 | 3.42 | 3.67 | 1.90 | 1.14 | 5 | 4 |
| | DBO-BP | 3.25 | 3.5 | 3 | 4 | 2.92 | 3.33 | 2.42 | 3.83 | 1.32 | 0.63 | 2 | 2 |
| | IDBO-BP | 3 | 3.5 | 2.83 | 4.25 | 3.08 | 3.75 | 2.67 | 4.08 | 1.31 | 0.54 | 1 | 1 |
| TH | BP | 4.08 | 1 | 4.75 | 2.08 | 4.17 | 1 | 4.33 | 2.25 | 1.88 | 2.17 | 4 | 6 |
| | TSSA-BP | 4.17 | 4.5 | 3.92 | 3 | 3.83 | 4.08 | 3.08 | 3.33 | 2.16 | 1.45 | 5 | 4 |
| | GWO-BP | 3.33 | 3.83 | 3 | 3.67 | 3.92 | 4.17 | 3.75 | 3.58 | 1.84 | 1.13 | 3 | 3 |
| | IGWO-BP | 3.83 | 4.42 | 3.58 | 3.08 | 4.08 | 4.5 | 4 | 3.17 | 2.27 | 1.54 | 6 | 5 |
| | DBO-BP | 3.42 | 3.67 | 3 | 4 | 2.58 | 3.42 | 2.42 | 3.83 | 1.34 | 0.60 | 2 | 2 |
| | IDBO-BP | 3.17 | 3.58 | 2.75 | 3.47 | 2.42 | 3.43 | 2.45 | 3.99 | 1.29 | 0.56 | 1 | 1 |

For comparison purposes, $R^2$ is taken as a negative value when calculating the average and overall rankings.

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
