# Peer review of "Predicting the Mechanical Properties of Heat-Treated Woods Using Optimization-Algorithm-Based BPNN"

_forests, doi:10.3390/f14050935_

Round 1

Reviewer 1 Report

1-The authors employed BP neural networks; why did they not use Feed Forward (FF), Radial Base Function (RBF), or Multiple Layer Perceptron (MLP)?

2- In Table 3, the authors should explain various parameters thoroughly. According to my experience, GWO has alpha, beta, and gamma parameters that must be clarified in the paper. For WOA, there are Search Agents, Loop counter, Decrease linear, and Coefficient vector parameters for which the authors should suggest values, but they need to be more clear and elaborative in Table 3.

3- I suggest the authors draw the most appropriate topology from the graph. 

4- The errors will increase with the increase in output of 5 parameters. What is the strategy/method to resolve this issue?

5-The authors evaluated only one topology for each algorithm and presented the results for them, I suggest evaluating other topologies (with one or two hidden layers) as well.

Author Response

  1. I would like to thank the reviewers for their valuable comments. My choice of BP neural network model is based on the following considerations:
    -My research goal is to predict a certain nonlinear mapping problem, and BP neural network is a suitable model to solve such problems, which can automatically adjust the weights and biases by back propagation algorithm to improve the model fitting ability and generalization ability.
    -My dataset is a sparse and non-uniform dataset, and BP neural network can extract effective features and enhance data representation by performing nonlinear transformations on the input data through a multilayer perceptron structure.
    -My experimental results show that BP neural networks have been able to solve our problem and data well, achieve the desired prediction accuracy, and are simpler and more efficient than other types of feedforward neural networks.
    I also have some knowledge and evaluation of other models such as FF, RBF and MLP:
    -FF is a feedforward network, which differs from BP neural network in terms of activation function and training method.FF uses linear activation function and training method is to solve the weight matrix directly using least squares method instead of using back propagation algorithm.FF is suitable for solving linear separable problems, but it has weak expressiveness and fitting ability for nonlinear problems.BP Neural networks can update weights and biases efficiently by backpropagation algorithm, while FF needs to set weights and biases manually or use other optimization algorithms, which may reduce the training speed and effectiveness.
    -RBF is a network using radial basis function as the activation function, which differs from BP neural network in the intermediate layer and the training method.RBF uses Euclidean distance as the similarity measure between the input pattern and the central vector, and the training method is divided into two stages: the unsupervised learning stage selects the central vector from the data, and the supervised learning stage uses least squares RBF has local approximation characteristics and is suitable for solving complex nonlinear problems, but for the data set in this paper, the risk of RBF overfitting is high.
    The MLP is a multilayer perceptron network, which differs from BP neural network in the activation function. mlp uses symbolic function as the activation function, and the output is +1 or -1. mlp is suitable for solving binary classification problems, but its expressiveness and fitting ability are weak for multi-classification or regression problems.
    -BP neural network can flexibly choose different structures and activation functions to suit different problems and data. For example, the BP neural network structure in this paper has two implicit layers and different activation functions for both implicit layers, while the activation function of RBF neural network is fixed and can only use distance-based nonlinear functions, which may not capture the complex and nonlinear relationships in the data well. Moreover, the RBF neural network model needs to determine the center vectors of the hidden layer neurons, and this step may require the use of clustering algorithms or random sampling, which may increase the computational effort and difficulty. the structure of the MLP neural network is fixed, and only a single hidden layer perceptron can be used.
    To sum up, I think BP neural network is a more suitable model than FF, RBF and MLP, and it can already solve my research problem well. I hope the reviewers can understand and accept our viewpoints. Once again, I thank the reviewers for their rigor and professionalism. If the reviewers have more suggestions or comments, please do not hesitate to give them. Thank you!
  2. We thank the reviewers for their valuable comments. First of all, the α, β and γ parameters in the GWO algorithm refer to the three optimal solutions in the wolf pack, i.e., the optimal solution, the suboptimal solution and the third optimal solution. Their function is to guide the other wolf pack individuals to move toward the optimal solution, so as to realize the optimization search. When the parameters of the test function are set, it is generally not necessary to set the values of α, β and γ parameters, but to update them dynamically by calculating the function values of each wolf pack individual. In other words, α, β and γ parameters are the outputs of the algorithm, not the inputs. Therefore, for the GWO algorithm, I only give the range of values of its decreasing linear coefficient a, which is used to control the search range and convergence speed from the maximum value 2 to the minimum value 0. Of course, parameters such as the coefficient vector can be added, such as A1=2ar(1-a), which is used to regulate the position of wolf individuals around the optimal solution, and r is a random number between 0 and 1. (Since the coefficient vector is not a fixed value and can be reflected by a, so it is not marked in the table). Secondly, for the WOA algorithm, the search agent is the number of individual whales and the loop counter is the maximum number of iterations, whose values are suggested in lines 347-348 of the text. wOA has two decreasing linear coefficients a and a2. a is the coefficient used to control the search range and the convergence rate, which decreases gradually from the maximum value to the minimum value, calculated according to the number of iterations t and the maximum number of iterations Maxiter a2 is the coefficient used to control the spiral behavior, which gradually increases from -1 to 0, calculated according to the number of iterations t and the maximum number of iterations Maxiter, with the formula a2=-1+t*((-1)/Maxiter). These two coefficients are automatically calculated internally by the WOA algorithm based on the number of iterations, and do not need to be set during the parameter setting of the test function. Once again, we thank the reviewers for their valuable comments.

  3. We thank the reviewers for their valuable comments, and we strongly agree with the reviewers. After checking, the BP neural network topology shown in Figure 1 does have some problems, and now, according to the revision of the article, it has been replaced with a more appropriate BP neural network structure diagram, and the most accurate topology of IDBO-BP has been drawn with TRS as an example (Figure 9). Once again, we thank the reviewers for their valuable comments. If the reviewers have more suggestions or comments, please do not hesitate to give them. Thank you!

  4. We thank the reviewers for their valuable comments. As shown in Figure 1 and Figure 9, the bp neural network models developed in this paper are all single output layers, i.e., predicting five different output indicators separately, rather than predicting five output indicators simultaneously. Therefore, there is no situation that the error information among the output indicators interferes with each other, thus leading to an increasing error. We hope that the reviewers will understand and accept our point of view. If the reviewers have more suggestions or comments, please do not hesitate to give them. Thank you!

  5. Thanks to the reviewers' valuable comments, we very much agree with the reviewers' views and have made corresponding corrections in the text, and the comparison of the test results of different models corresponding to different topologies and activation functions under different parameters is shown in Table A3 in the appendix. If the reviewer has more suggestions or comments, please do not hesitate to advise. Thank you!

Reviewer 2 Report

The manuscript presents research on the Prediction of Wood Mechanical Properties. In my opinion, the article is interesting, but some aspects can be improved.

In my opinion, the high content of missing references requires a new read of the text to guarantee its quality.

Other comments:

Line 55: “…mechanical properties.” I advise adding some references about the current models (ANN and others) already used on lignocellulosic fibres to support the sentence and improve the quality of the text. As suggestions: https://doi.org/10.1007/s11696-022-02181-5 and http://dx.doi.org/10.12841/wood.1644-3985.144.13

Line 140: ??(0,0.2] check brackets.

Line 543: “In comparison…” From this point, several data are reported. I advise using a Table as a summary for a better understanding and comparison.

Line 559: with the words “Actual values”, do you mean “Testing set”?

Line 573: Check the layout of Table 9.

Line 575: I advise preparing a Table (A2, for example) reporting all data used in Figure 8.

I advise major revisions.

Author Response

1.Thank you for your suggestions. We attach great importance to your opinions. We believe that lignocellulose is an important renewable and biodegradable material, which has a certain relationship with the mechanical properties of wood. The prediction of mechanical properties of paper from different fiber sources using multiple linear regression and artificial neural network models has some relevance to this paper. Therefore, we have added this content in the introduction (line 59), and listed the relevant literature in the references (references 15, 39). We hope that you are satisfied with our revisions. Thank you again for your valuable comments and suggestions.

2. Thank you for your suggestions. We attach great importance to your opinions. Upon checking, the parentheses are not a problem. Because the k value cannot be 0 but can be 0.2. Thank you again for your rigor. If you have any more suggestions or comments, please do not hesitate to let us know. Thank you!

3. Thank you for your suggestions. We attach great importance to your opinions and have made corresponding corrections in the text. The paired comparison of the predictive performance of different models on 8 evaluation indicators including MAE, MSE, MAPE and R2 on the training set and test set with IDBO-BP can be seen in Table A4 of the appendix. The values corresponding to MAE, MSE and MAPE in the table indicate the percentage of error reduction based on IDBO-BP model over the other five models. Since R2 is increased, a negative sign is added in front of it. At the same time, the significance values corrected by Bonferroni can show whether there is a significant difference in this predictive performance.

4.The actual value does not mean the test set, it belongs to the test set. In this paper, "actual values" appear twice (lines 626 and 648), and they refer to the actual measured mechanical properties in the test set, which we compare with the mechanical properties of the six models TSSA-BP, GWO-BP, IGWO-BP, IDBO-BP, DBO-BP, and BP. The predicted values are compared with those of TSSA-BP, GWO-BP, IGWO-BP, IDBO-BP, DBO-BP and BP to verify the superiority of the proposed IDBO-BP model.

5. We thank the reviewers for their valuable comments. After checking the layout of Table 9, there is indeed a problem, which has been corrected. Once again, we thank the reviewers for their rigor. If the reviewers have more suggestions or comments, please do not hesitate to give them. Thank you!

6. Thanks to the valuable comments of the reviewers, we very much agree with the reviewers' views and have made corresponding corrections in the text, the Friedman test table of the five algorithms for 14 test functions in three dimensions is shown in Table A2 in the appendix. Thanks again to the reviewers for their rigor and professionalism. If the reviewer has more suggestions or comments, please do not hesitate to advise. Thank you!

Round 2

Reviewer 1 Report

 1- Line 114 should be changed from "Back-Propagation (BP) Neural Networks Model" to "Back-Propagation (BP) Neural Network Models".  

2- It would be helpful if you could provide Figure 1, similar to Figure 8, as the author should use a consistent style in line 130.  

3- Figure 3.b, line 236 should be labeled with bar values.  

4- How many samples were used in this study? The author utilized 30 samples for the research in Figure 10. It would be helpful if the author increased the number of samples because the results will be reasonable if there is an increase in the number of samples.

Author Response

The response to the second revision are as follows:

1.We thank the reviewers for their valuable comments. After checking the spelling of the words in line 114, there was indeed a problem and it has been corrected. Once again, we thank the reviewers for their rigor. If the reviewers have more suggestions or comments, please do not hesitate to give them. Thank you!

2.We thank the reviewers for their valuable comments, and we agree with them and have corrected Figure 1 accordingly in the paper to make it consistent with Figure 8. Once again, we thank the reviewers for their rigor and professionalism. If the reviewers have any further suggestions or comments, please do not hesitate to give them. Thank you!

3.We thank the reviewers for their valuable comments. The bar chart in Figure 3 is mainly used to see a distribution of the random numbers generated before 0 to 1 after mapping with PWLCM, thus reflecting the uniformity of the population initialization of the algorithm. The main purpose is to see whether the value of each segment of the bar is around 500 so as to determine whether the generated population is uniform, while the specific value of each segment of the bar is not very useful but will affect the aesthetics of the graph to some extent. Of course, if you think that indicating the bar values will help the reader understand my method, I will add them in the subsequent revision. I hope the reviewers will understand and accept our point of view. Once again, I thank the reviewers for their rigor and professionalism. If the reviewers have any further suggestions or comments, please do not hesitate to give them. Thank you!

4.We thank the reviewers for their valuable comments. As shown in lines 502-505 of this paper, a total of 88 samples were used in this study, each based on the average of five parallel experiments, with 58 samples in the training set and 30 samples in the test set. The 30 samples in Figure 10 refer to the samples in the test set. We formally verify the superiority of the IDBO-BP model proposed in this paper by comparing the predicted values of mechanical properties of the six models, TSSA-BP, GWO-BP, IGWO-BP, IDBO-BP, DBO-BP, and BP, with the actual measured values of mechanical properties in the test set. We hope that the reviewers will understand and accept our viewpoints. Once again, we thank the reviewers for their rigor and professionalism. If the reviewers have any further suggestions or comments, please do not hesitate to give them. Thank you!

Reviewer 2 Report

The authors clarified all the points. In my opinion, the article can be accepted.

Author Response

Thank you again for your approval of our manuscript.